# Coupling of H3K27me3 recognition with transcriptional repression through the BAH-PHD-CPL2 complex in *Arabidopsis*

Yi-Zhe Zhang [1,2,6], Jianlong Yuan [1,2,6], Lingrui Zhang[3,6], Chunxiang Chen[1], Yuhua Wang[1], Guiping Zhang[1], Li Peng[1], Si-Si Xie [1,2], Jing Jiang[4], Jian-Kang Zhu [1,3✉], Jiamu Du [5✉] & Cheng-Guo Duan [1,4✉]

Histone 3 Lys 27 trimethylation (H3K27me3)-mediated epigenetic silencing plays a critical role in multiple biological processes. However, the H3K27me3 recognition and transcriptional repression mechanisms are only partially understood. Here, we report a mechanism for H3K27me3 recognition and transcriptional repression. Our structural and biochemical data showed that the BAH domain protein AIPP3 and the PHD proteins AIPP2 and PAIPP2 cooperate to read H3K27me3 and unmodified H3K4 histone marks, respectively, in *Arabidopsis*. The BAH-PHD bivalent histone reader complex silences a substantial subset of H3K27me3-enriched loci, including a number of development and stress response-related genes such as the RNA silencing effector gene *ARGONAUTE 5* (*AGO5*). We found that the BAH-PHD module associates with CPL2, a plant-specific Pol II carboxyl terminal domain (CTD) phosphatase, to form the BAH-PHD-CPL2 complex (BPC) for transcriptional repression. The BPC complex represses transcription through CPL2-mediated CTD dephosphorylation, thereby causing inhibition of Pol II release from the transcriptional start site. Our work reveals a mechanism coupling H3K27me3 recognition with transcriptional repression through the alteration of Pol II phosphorylation states, thereby contributing to our understanding of the mechanism of H3K27me3-dependent silencing.

[1] Shanghai Center for Plant Stress Biology and CAS Center for Excellence in Molecular Plant Sciences, Chinese Academy of Sciences, 201602 Shanghai, China. [2] University of Chinese Academy of Sciences, 100049 Beijing, China. [3] Department of Horticulture and Landscape Architecture, Purdue University, West Lafayette, IN 47907, USA. [4] State Key Laboratory of Crop Stress Adaptation and Improvement, School of Life Sciences, Henan University, 475004 Kaifeng, China. [5] Key Laboratory of Molecular Design for Plant Cell Factory of Guangdong Higher Education Institutes, Institute of Plant and Food Science, School of Life Sciences, Southern University of Science and Technology, 518055 Shenzhen, China. [6] These authors contributed equally: Yi-Zhe Zhang, Jianlong Yuan, Lingrui Zhang. ✉email: jkzhu@psc.ac.cn; dujm@sustech.edu.cn; cgduan@psc.ac.cn

  1

In eukaryotic cells, the N-terminal histone tails undergo numerous posttranslational modifications (PTMs). Histone modification acts as a mark to specify the chromatin status, as well as potential functional indications[1]. The deposition, recognition, and removal of specific histone PTMs are dynamically regulated by different proteins or protein complexes called "writer", "reader", and "eraser" modules, respectively[2–4]. The reader module can specifically recognize certain histone mark in both sequence- and modification-specific manners, and subsequently transmits the signal to downstream effectors. As a repressive epigenetic mark localized in euchromatin, deposition of trimethylation on histone H3 lysine 27 (H3K27me3) has been observed in many important functional genes[5]. The polycomb repressive complexes (PRCs), consisting of a different polycomb group (PcG) of proteins, have been shown to be involved in the deposition and downstream action of H3K27me3 mark[6]. Different PcG proteins associate to form two functionally distinct complexes, PRC1 and PRC2. The PRC1 complex has E3 ligase activity which has been shown to catalyze the monoubiquitination of histone H2A at lysine (H2Aub1), and the PRC2 complex catalyzes H3K27me2 and H3K27me3 (refs. [7–10]). Three major models have been proposed to explain the mechanisms of PRC complex-mediated transcription repression[8]. For those bivalent promoters marked by both H3K27me3 and H3K4me3 marks, PcG complexes are believed to hold the poised Pol II at the transcription start site (TSS), resulting in the inhibition of Pol II release. Alternatively, PcG complexes can alter the chromatin environment by inducing chromatin condensation, thereby blocking the accessibility of chromatin remodeling complexes that is required for transcription activation[11–13]. Third, the histone PTMs might directly prevent Pol II processivity during transcription elongation[8]. For example, studies in *Drosophila* have indicated that H3K27me3 could limit Pol II recruitment to gene promoters[14]. H2Aub1 has been implicated in restraining Pol II elongation[15,16]. However, the detailed mechanisms through which H3K27me3 reading is connected to transcriptional repression are not fully understood.

Histone mark recognition in plants is generally similar to that in animals, but sometimes possesses plant-specific mark–reader pairs[17]. The PHD and BAH domains are two types of histone-binding domains in eukaryotes[18,19]. The PHD finger has been reported to recognize methylated/unmethylated H3K4 marks and lysine acetylation marks[19], and the BAH domain can bind distinct histone marks, including H3K9me2 (ref. [20]), H4K20me2 (ref. [21]), unmodified H3K4 (ref. [22]), nucleosome core particle[23–26], and the more recently identified H3K27me3 (ref. [27–30]). In plants and animals, a large number of development and environmental response-related processes are subjected to H3K27me3-dependent regulation. Among them, flowering control has been a paradigmatic model for PRC complexes-mediated transcriptional repression in plants. The H3K27me3 dynamics in the flowering repressor gene *FLOWERING LOCUS C* (*FLC*) and the florigen gene *FLOWERING LOCUS T* (*FT*) play essential roles in the flowering time control[31], and the H3K27me3 regulators influence flowering time in different ways[27–29,32–40]. Here, we demonstrated that the BAH domain-containing protein anti-silencing 1 (ASI1)-IMMUNOPRECIPITATED PROTEIN 3 (AIPP3) and two PHD domain-containing proteins AIPP2/ PARALOG OF AIPP2 (PAIPP2) could form a BAH–PHD module to read H3K27me3 and unmethylated H3K4, respectively, and coordinate in implementing transcriptional repression of hundreds of genes, particularly those development and stress-responsive genes in *Arabidopsis*, such as the florigen gene *FT* and the RNA silencing effector gene *AGO5*. Moreover, our structural and biochemical studies further revealed the molecular basis for the specific recognition of these histone marks. We also revealed

that the BAH–PHD module represses the release of Pol II from TSS regions by cooperating with CPL2, a known plant-specific Pol II carboxyl terminal domain (CTD)-Ser5 phosphatase. Collectively, our findings reveal a coupling of the H3K27me3 recognition and downstream transcriptional repression through the BPC complex. This pathway may represent a mechanism of H3K27me3-mediated gene silencing.

## Results

**BAH protein AIPP3 associates with PHD proteins and CPL2 to form a protein complex in *Arabidopsis*.** Through a mass spectrometry (MS) analysis of the chromatin regulator ASI1, we previously demonstrated that BAH domain-containing protein AIPP3, and PHD protein AIPP2 and CPL2 are associated with ASI1 (refs. [41,42]). AIPP2 is known to interact with AIPP3 and CPL2 (ref. [41]). This association was further confirmed by immunoprecipitation assays coupled to a mass spectrometry analysis (IP–MS) of AIPP3, AIPP2, and CPL2 in which AIPP3, AIPP2, and CPL2 could be mutually co-purified with one another except for ASI1 (Fig. 1a and Supplementary Data 1). Interestingly, another PHD protein encoded by *AT5G16680* (Supplementary Fig. 1), the closest paralog of AIPP2 (hereafter referred to as PAIPP2) in *Arabidopsis*, was co-purified with AIPP3 and CPL2, and AIPP3 and CPL2 were also present in the IP–MS of PAIPP2 (Fig. 1a). The yeast two-hybrid (Y2H) and split luciferase assays indicated that PAIPP2 could also interact with AIPP3 and CPL2, but not with AIPP2 (Fig. 1b and Supplementary Fig. 2). Regarding the domain requirements for protein interactions, the Y2H results indicated that the BAH domain-containing N-terminus of AIPP3 and the RBM motif-containing C-terminus region of CPL2 are required for their interactions with AIPP2 and PAIPP2, respectively (Supplementary Fig. 3). AIPP2 and PAIPP2 were divided into three parts: the N-terminus (N), PHD that is followed by a frequently associated polybasic region (PHD–PBR), and the C-terminus (C) (Supplementary Fig. 4). The PHD–PBR part is indispensable for AIPP2/PAIPP2–AIPP3 interactions. Intriguingly, the PHD–PBR interaction with AIPP3 could be strengthened, and inhibited by the N and C termini of AIPP2/ PAIPP2, respectively (Supplementary Fig. 4), indicating the presence of an intramolecular regulation mode within AIPP2 and PAIPP2. The C termini of AIPP2 and PAIPP2 were fully responsible for the interaction with CPL2. Thus, we reasoned that the PHD proteins AIPP2 and PAIPP2 interact with AIPP3 and CPL2 independently to associate in vivo (Fig. 1c). Moreover, the gel filtration assay that was performed using epitope-tagged transgenic lines indicated that these four proteins co-eluted in the same fractions (Fig. 1d). Thus, these data support the notion that the BAH protein AIPP3 associates with two PHD proteins and CPL2 to form a protein complex in vivo (which are hereafter referred to as the BAH–PHD–CPL2 complex or the BPC complex).

**The BPC complex represses flowering by inhibiting *FT* expression.** By generating native promoter-driven β-glucuronidase (*GUS*) reporter transgenes, we observed that the BPC complex genes were ubiquitously expressed in *Arabidopsis* (Supplementary Fig. 5). To explore the biological function of the BPC complex, the null mutants of *PAIPP2* were generated using CRISPR/Cas9-mediated gene editing (Supplementary Fig. 6), and the morphology and flowering phenotypes of the generated *paipp2-1* mutant, as well as the reported *aipp3-1*, *aipp2-1*, and *cpl2-2* mutants were investigated[41]. The *aipp3-1*, *cpl2-2*, and *aipp2-1/paipp2-1* displayed multiple developmental defects, such as a dwarfed size, small leaves, and poor fertility (Fig. 2a). Instead, the *aipp2-1* and *paipp2-1* mutants only showed mild

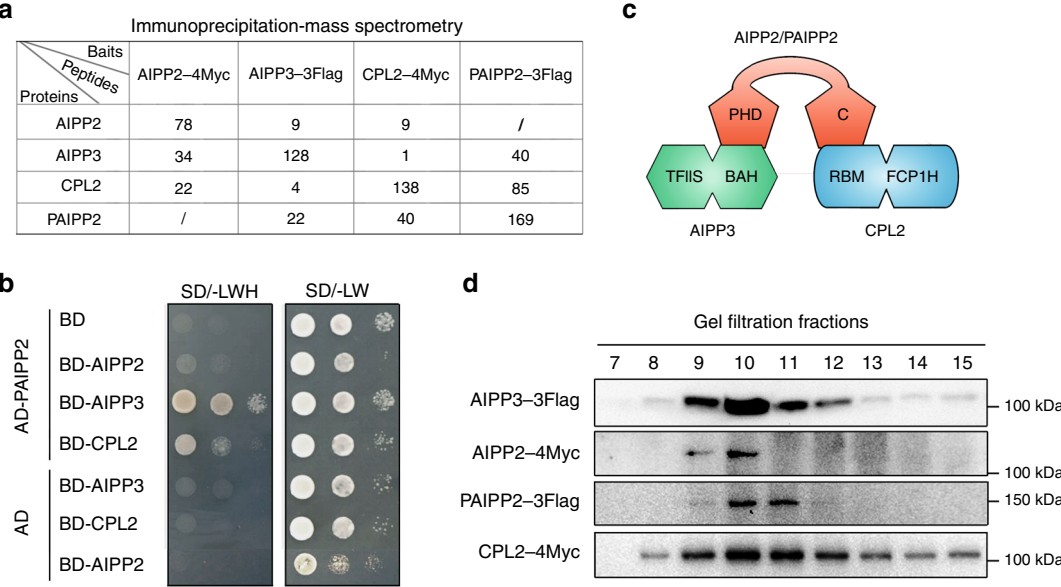

**Fig. 1 The BAH protein AIPP3 associates with PHD proteins and CPL2 to form a protein complex. a** Mass spectrometry analysis of epitope-tagged *AIPP3, AIPP2, PAIPP2,* and *CPL2* transgenic plants. **b** Y2H results showing the reciprocal interactions within the tested proteins. **c** A diagram showing the interaction network within AIPP3, AIPP2, PAIPP2, and CPL2. **d** Immunoblotting results of gel filtration assays. One of three independent experiments was shown.

developmental defects. Moreover, the *aipp3-1* and *cpl2-2* mutants showed obvious earlier flowering during both long day (LD) and short day (SD) photoperiods in comparison with Col-0 (Fig. 2b, c). Although only mild early flowering was observed in the *aipp2-1* and *paipp2-1* single mutants, *aipp2-1/paipp2-1* showed similar early flowering compared with the *aipp3-1* and *cpl2-2* mutants (Fig. 2b, c), suggesting a redundancy in these two PHD proteins in relation to the flowering time control. To dissect their genetic relationship, we attempted to generate double, triple, and quadruple mutants. Unfortunately, we failed to obtain the *aipp3/aipp2/paipp2/cpl2* quadruple mutant due to a severe developmental defect. Compared to Col-0, the *aipp3-1/aipp2-1/paipp2-1, aipp3-1/cpl2-2,* and *aipp2-1/paipp2-1/cpl2-2* mutants flowered earlier, and the time to flower was similar to the single mutants under the LD condition (Fig. 2d), suggesting that the BPC complex acts in the same genetic pathway in flowering time control.

In *Arabidopsis*, FLC, which is a MADS-box transcription factor that integrates multiple flowering signals, acts as a key floral repressor[31,43,44]. FLC directly represses the expression of florigen gene *FT* and *SUPPRESSOR OF OVEREXPRESSION OF CO 1* (*SOC1*) by binding to the promoter of *SOC1* and the first intron of *FT*[45]. The quantitative RT-PCR (RT-qPCR) indicated that the *FLC* RNA levels were reduced in the *aipp3-1, aipp2-1, cpl2-2, paipp2-1,* and *aipp2-1/paipp2-1* mutants compared to Col-0 (Fig. 2e). Instead, the *FT* RNA levels were significantly increased in the *aipp3-1, cpl2-2,* and *aipp2-1/paipp2-1* mutants, but not in the *aipp2-1* and *paipp2-1* single mutants, further supporting the functional redundancy of AIPP2 and PAIPP2. By contrast, the *SOC1* RNA level was not significantly changed in all the tested mutants. It is known that the accumulation of FT protein has a circadian rhythm that peaks before dusk during LD photoperiod[46-48]. We noticed that the loss of the BPC complex did not change the circadian rhythm of *FT* mRNA, but led to a constitutive increase (Fig. 2f). To determine the genetic relationship between FLC, FT, and the BPC complex in relation to the flowering time control, *flc-3,* and *ft-10* mutants (in Col-0 background) were crossed with the tested mutants. Surprisingly, the *aipp3-1/flc-3, cpl2-2/flc-3,* and *aipp2-1/paipp2-1/flc-3* mutants displayed similar early flowering compared with the *aipp3-1,*

*cpl2-2,* and *aipp2-1/paipp2-1* mutants, but they flowered earlier than the *flc-3* single mutant (Fig. 2g). By contrast, the early flowering phenotypes of *aipp3-1, cpl2-2,* and *aipp2-1/paipp2-1* mutants were completely rescued by *ft-10* (Fig. 2h), indicating that the BPC complex represses flowering primarily by repressing the expression of *FT*.

**The AIPP3-BAH domain specifically recognizes the H3K27me3 mark.** The chromatin-based mechanisms play vital roles in the flowering time control[1,49,50]. The BAH domain is commonly identified as an epigenetic reader module of a particular histone mark[18]. To decipher the molecular function of the AIPP3-BAH domain (Fig. 3a), we firstly performed a histone peptide pull-down assay using purified AIPP3-BAH protein. Among all the tested histone peptides, AIPP3-BAH could only be pulled down by H3K27me1, H3K27me2, and H3K27me3 peptides in a sequentially increasing manner (Fig. 3b). The H3K27me3-binding activity was further confirmed by isothermal titration calorimetry (ITC) binding analysis (Fig. 3c). Among all four tested histone methylation marks, H3K4me3, H3K9me2, H3K27me3, and H3K36me3, the AIPP3-BAH domain showed a significant preference for the H3K27me3 mark. Moreover, consistent with the histone pull-down result, AIPP3-BAH has a preference for the higher methylation level in H3K27 (Fig. 3c). Thus, the evidence above fully demonstrated that AIPP3-BAH is a H3K27me3-reader module.

**Structure of the AIPP3-BAH domain in complex with an H3K27me3 peptide.** To gain molecular insight into the interaction between the AIPP3-BAH domain and H3K27me3, we successfully determined the crystal structure of the AIPP3-BAH domain in complex with an H3K27me3 peptide at a resolution of 2.4 Å (Fig. 3d and Supplementary Table 1). Overall, the AIPP3-BAH domain has a classic β-barrel structure that is similar to other reported BAH domain structures[18]. The H3K27me3 peptide has a good electron density map and a β-strand-like extended conformation (Supplementary Fig. 7a). The peptide is captured by a negatively charged cavity that is formed on the surface of the

AIPP3-BAH domain with extensive hydrophobic and hydrophilic interactions (Fig. 3e). In detail, the three aromatic residues of the AIPP3-BAH domain, Tyr149, Trp170, and Tyr172, form an aromatic cage to accommodate the trimethyl-lysine of the H3K27me3 peptide (Fig. 3f), and it resembles other typical methyl-lysine-reading histone readers[51]. At the N-terminus, the H3A25 forms two main chain–main chain hydrogen bonds with

the AIPP3 Val140 and Glu142 (Fig. 3f). In the middle, the AIPP3 residues Trp170, Tyr172, His198, and Asp200 form an extensive hydrogen-binding network with H3S28, which further fixes the imidazole ring of His198 in a special rotamer state that is parallel with the prolyl ring of H3P30. This parallel alignment of the two rings enables the CH-π, and stacking interactions between AIPP3 His198 and H3P30, which resemble the recognition of

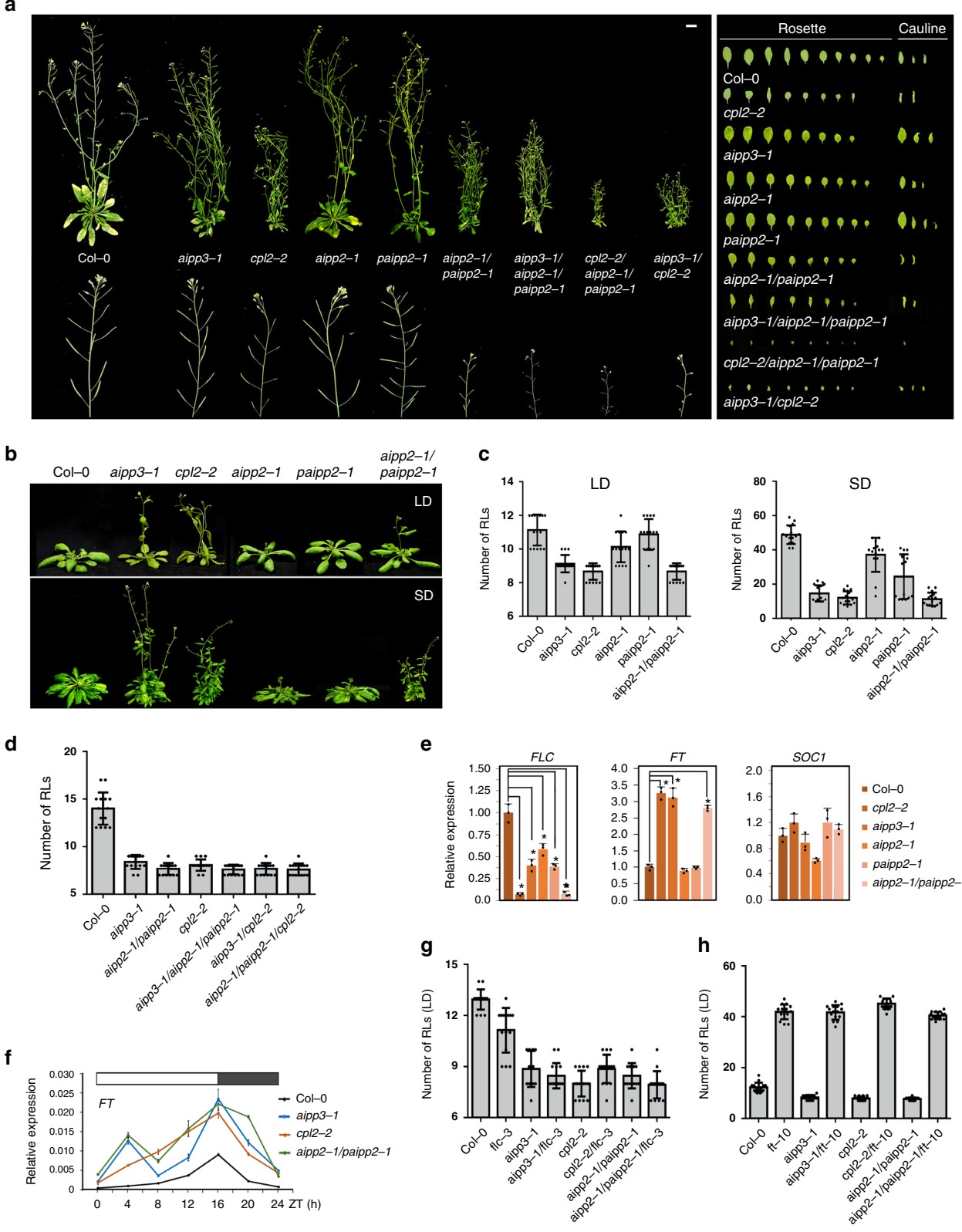

**Fig. 2 The BAH–PHD–CPL2 complex regulates plant development and flowering time. a** The phenotypic developmental defects of *bpc* mutants. The morphological phenotypes of whole plants, inflorescence tissues, rosette, and cauline leaves were shown. **b, c** The flowering phenotypes (**b**) and the numbers of rosette leave (RLs) at flowering (**c**) in selected mutants during LD and SD photoperiods. Black horizontal lines represent the mean, and the error bars represent ±S.D. from the number of plants counted for each genotype. Col-0: *n* = 15 (LD/SD), *aipp3-1*: *n* = 15 (LD/SD), *cpl2-2*: *n* = 15 (LD/SD), *aipp2-1*: *n* = 15/14 (LD/SD), *paipp2-1*: *n* = 15/14 (LD/SD), aipp2-1/paipp2-1: *n* = 15 (LD/SD). **d** Comparison of the number of RLs at flowering in selected mutants under the LD condition. Black horizontal lines represent the mean, and the error bars represent ±S.D. from the number of plants counted for each genotype. *n* = 15 per line. **e** The relative mRNA levels of the *FLC*, *FT*, and *SOC1* genes in the selected mutants. The mRNA levels were first normalized to *ACT2* and then to Col-0. The data are the means ± S.D. of three biological repeats. Unpaired one-tailed *t* test was performed and *\*p* value < 0.01. **f** The circadian accumulation of *FT* mRNA in selected mutants. The *FT* mRNA levels were normalized to *ACT2*. The data are the means ± S.D. of three biological repeats. The white and black boxes represent light and dark periods, respectively. **g, h** The numbers of RLs in the BAH–PHD–CPL2 complex mutants and their double mutants with *flc-3* (**g**) and *ft-10* (**h**) at flowering during the LD photoperiod. Black horizontal lines represent the mean, and the error bars represent ±S.D. from the number of plants counted for each genotype. *n* = 15 per line.

H3K27me3 by the EBS and SHL BAH domains with conserved key residues (Supplementary Fig. 7b)[27,28]. In the C-terminus, H3G33 interacts with AIPP3 Val240 through a main chain–main chain hydrogen bond. To validate these structural observations, we performed a mutagenesis analysis on the key residues. The mutations D200A and H198A, which are essential for H3P30 recognition, in addition to Y149A, W170A, and Y172A, which represent important aromatic cage residues, showed reduced or no detectable binding to the H3K27me3 peptide (Fig. 3g, h).

**PHD fingers of AIPP2 and PAIPP2 recognize the unmodified H3K4 mark.** In addition to AIPP3-BAH, the PHD fingers of AIPP2 and PAIPP2 may also be involved in the recognition of histone marks. AIPP2 and PAIPP2 share a conserved PHD finger in their sequences (Supplementary Fig. 8a), and their sequences of PHD fingers possess the typical signatures of unmodified H3 recognition PHD fingers[19]. We first detected their histone substrate binding properties by ITC method. Although the AIPP2-PHD finger does not behave well in vitro and tends to precipitate, we successfully detected the binding between the PAIPP2-PHD finger and the differentially methylated H3K4 peptides (Fig. 3i). The PAIPP2-PHD finger prefers to bind to unmethylated H3K4 and the binding affinities were clearly decreased when methylation level of the H3K4 peptide was increased. Considering the high sequence similarity between the two PHD fingers (Supplementary Fig. 8a), we believe that the AIPP2-PHD finger may possess the same binding preference on the unmodified H3K4. Moreover, both AIPP2 and PAIPP2-PHD fingers share ~35% sequence identity with the PHD finger of *Glycine max* ATXR5 (PDB code: 5VAB; Supplementary Fig. 8a), which recognizes the unmodified H3K4 (ref. [52]). We modeled the AIPP2 and PAIPP2-PHD fingers using the ATXR5 PHD finger, as a template to analyze the interactions with unmodified H3K4 (Fig. 3j and Supplementary Fig. 8b). In the modeled structure, we noticed that almost all the peptide-binding residues are conserved (Supplementary Fig. 8a). For instance, the Pro322 and Gly324 of PAIPP2, which correspond to the ATXR5 Pro60 and Gly62, respectively, are involved in the hydrogen-bonding interaction with H3A1 (Fig. 3k). Similarly, the Ile300, Cys301, and Asp306 of PAIPP2, which are equivalent to ATXR5 Leu39, Cys40, and Asp44, respectively, contribute to the recognition of H3R2 (Fig. 3k). The Val285, Gly292, and Leu298 of PAIPP2, which correspond to the Val23, Gly31, and Leu37 of ATXR5, respectively, participate in the recognition of the unmodified H3K4 (Fig. 3k). To validate the modeling results, we performed mutagenesis experiment, too. As most of the peptide recognition is achieved by the hydrogen-bonding interactions of the main chain of the PHD finger, we only mutated Asp306 of PAIPP2, which is involved in the recognition of H3R2 by its side chain. As shown in Fig. 3l, the D306K mutation of PAIPP2-PHD finger almost totally disrupts the peptide binding, further supporting our

modeling data. The AIPP2-PHD finger possesses similar unmodified H3K4 recognition residues and interactions (Supplementary Fig. 8c).

**The BPC complex represses the expression of the genes marked by H3K27me3 and low-methylated H3K4.** H3K27me3 is usually considered a repressive mark that functions in transcriptional repression by recruiting recognition proteins or protein complexes[4,8]. To dissect the molecular function of the BPC complex, mRNA-seq was performed using their null mutants, as well as CLF and LHP1 mutants[53,54], which are one of the known H3K27me3 methyltransferases and a reader protein in *Arabidopsis*, respectively. Using a twofold change cutoff, we noticed that the numbers of upregulated differentially expressed genes (up-DEGs) were far greater than the numbers of down-DEGs in the *aipp3-1*, *cpl2-2*, and *aipp2-1/paipp2-1* mutants (Fig. 4a). Supporting the functional redundancy of AIPP2 and PAIPP2, very few DEGs were identified in *aipp2-1* and *paipp2-1* single mutants. It is noteworthy that most of the up-DEG genes regulated by the BPC complex display very low-expression levels in the wild type, which is a similar pattern to that the *clf* and *lhp1* mutants (Fig. 4b). Under a strict criterion, 155 genes were commonly upregulated in the *bah–phd–cpl2* mutants (Figs. 4c, d and Supplementary Data 2), and a substantial subset are stress-responsive genes (Supplementary Fig. 9), such as *SEC31A* (*AT1G18830*), which participates in endoplasmic reticulum stress responses[55–57], and *AGO5* (*AT2G27880*), which is involved in antiviral RNA silencing[58–60] and gametophyte development[61,62]. In fact, for most of the *aipp2-1/paipp2-1* up-DEGs, higher expression was also observed in the *aipp3-1* and *cpl2-2* mutants (Fig. 4d), indicating that the BPC complex targets a common subset of genes for repression.

We next explored the chromatin feature of the target genes by plotting the distributions of the histone marks on the up-DEGs, using published histone modifications chromatin immunoprecipitation (ChIP)-seq data[63]. As expected, the H3K27me3 mark was heavily enriched on the bodies of the common target genes (Fig. 4e). Consistent with the ITC result in which the higher methylation of H3K4 inhibits the binding of PAIPP2-PHD (Fig. 3i), the levels of active H3K4me3 mark were lower in the regions around the TSS. H3K36me3 deposition was also low on the whole gene bodies. The depositions of H3K9me2 and H3K27me1 were markedly lower on the target genes, suggesting that the BPC complex primarily targets euchromatic genes. The distribution patterns of H3K27me3 and H3K4me3 marks at BPC complex target genes were confirmed by ChIP-qPCR assays at representative target genes *AGO5*, *SUC5*, and *FT* (Fig. 4f, g). We next investigated the impacts of BPC complex dysfunction on global histone mark levels. Interestingly, compared to the reduction of global H3K27me2/3 levels in *clf-81* mutant, the absence of the BPC complex did not result in obvious changes in

global levels of H3K27me1/2/3 and H3K4me1/2/3 (Supplementary Fig. 10a). Consistent with this pattern, H3K27me3 deposition was not obviously changed between Col-0 and *bpc* mutants at selected target genes *AGO5*, *SUC5*, and *FT* (Fig. 4g), whereas H3K4me3 levels are slightly increased (Fig. 4h). To further verify this observation, more target genes were selected. As shown in Supplementary Fig. 10b–d, nuclear run-on assay showed that *AT2G43570*, *AT3G59480*, and *AT3G53650* genes were transcriptionally upregulated compared to wild type, indicating that these three genes are subjected to BPC complex-mediated

transcriptional repression. Interestingly, ChIP-qPCR results indicated that H3K27me3 levels were not obviously changed in *bpc* mutants at *AT2G43570* and *AT3G59480*, consistent with our observation at *AGO5*, *SUC5*, and *FT* (Fig. 4g). While H3K27me3 deposition showed significant reduction at *AT3G53650*, implying that BPC dysfunction has substantial impacts on H3K27me3 deposition at this gene. Combined with these data, we speculate that, for most target genes, the BPC complex may serve as a surveillance system to prevent reactivation of H3K27me3-marked genes, which are already silenced by PRC2, but in some specific

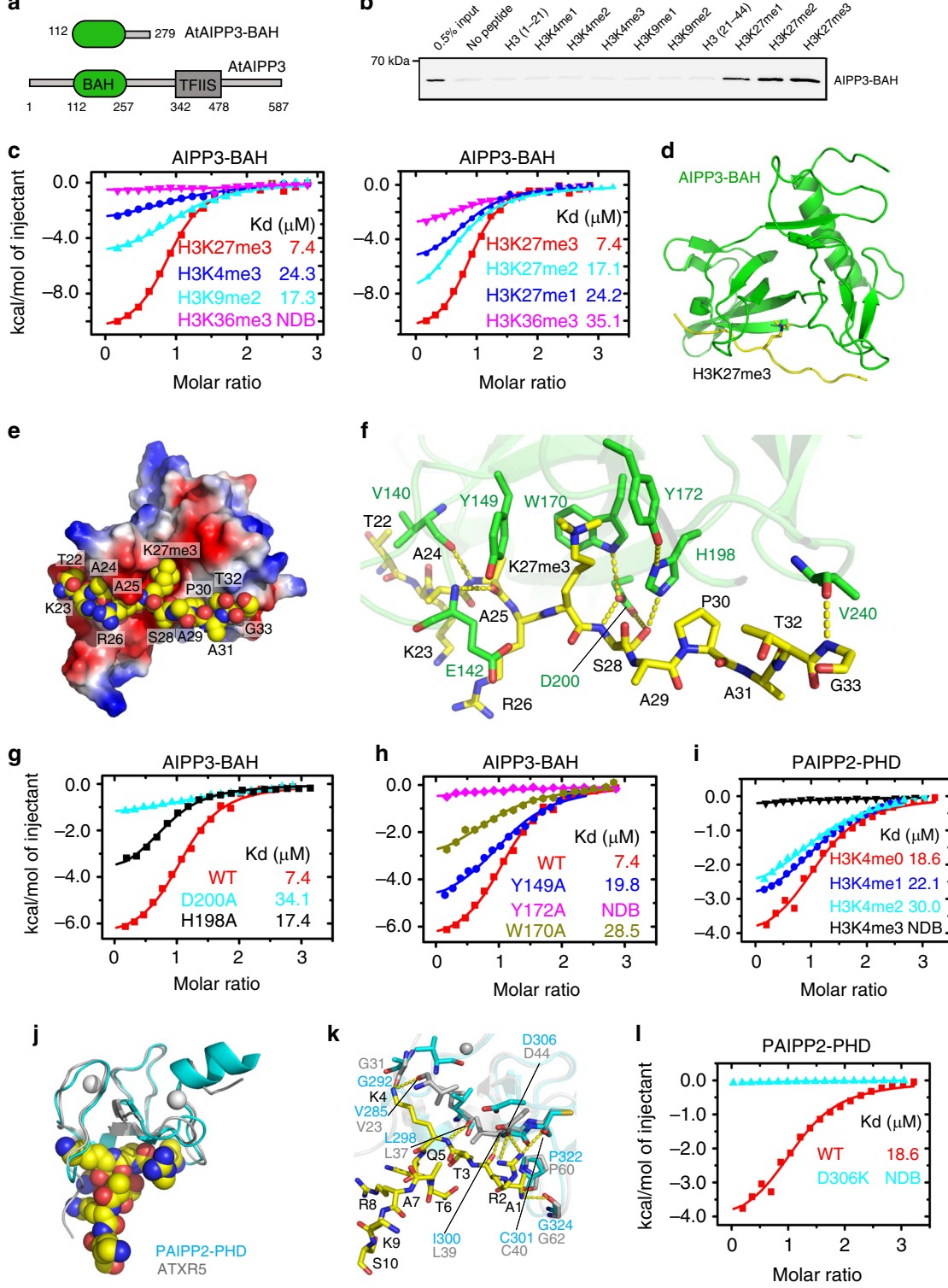

**Fig. 3 Structural analysis of the AIPP3-BAH domain and PAIPP2-PHD finger in complex with H3K27me3 and unmodified H3K4 peptide, respectively. a** The domain architecture of AIPP3 (lower panel) and the BAH domain construct used for structural and biochemical studies (upper panel). **b** The immunoblotting results showing the pull-down results for AIPP3-BAH using different histone peptides. The 0.5% inputs serve as positive controls. **c** The ITC binding curves showing the AIPP3-BAH domain-binding preference for different histone methylation marks (left panel) and different H3K27 methylation levels (right panel). NDB no detectable binding. **d** The overall structure of the AIPP3-BAH domain H3K27me3 peptide complex shown in ribbon with the AIPP3-BAH domain, and the H3K27me3 peptide colored in green and yellow, respectively. **e** An electrostatic surface view of AIPP3-BAH domain with the H3K27me3 peptide in the space-filling model showing that the peptide fits inside a negatively charged surface cleft of the AIPP3-BAH domain. **f** The detailed interaction between AIPP3-BAH domain and the H3K27me3 peptide with the interacting residues highlighted in sticks and the hydrogen bonds highlighted in dashed yellow lines. **g**, **h** The ITC binding curves show that the mutations of essential residues for the H3P30 recognition (**g**) and aromatic cage residues (**h**) of the AIPP3-BAH domain significantly decrease the binding toward the H3K27me3 peptide. **i** The ITC binding curves showing the specific preference of the PAIPP2-PHD finger for the unmodified H3K4. **j** The overall modeled structure of the PAIPP2-PHD finger in complex with the unmodified H3 peptide with the PHD finger and peptide shown in ribbon and space-filling models. The modeled PAIPP2-PHD finger and the modeling template ATXR5 PHD finger are colored in cyan and silver, respectively, and they were superimposed together. **k** The detailed interaction between the PAIPP2-PHD finger (in cyan) and the unmodified H3K4 peptide with the interacting residues are highlighted in stick model and hydrogen bonds highlighted in dashed yellow lines. The corresponding residues from modeling template ATXR5 (in silver, PDB code: 5VAB) were overlain and highlighted, showing that almost all the interacting residues are conserved. **l** The ITC binding curves showing that the D306K mutation of PAIPP2-PHD finger, which potentially disrupts H3R2 recognition, totally abolishes the unmodified H3K4 binding by PAIPP2.

---

genes, BPC is required for H3K27me3 deposition through unknown mechanism.

**The BPC complex reads the H3K27me3 at a genome-wide level**. We next investigated the interplay between the BPC complex and the chromatin of the target genes by performing a ChIP assay in epitope-tagged transgenic plants. The ChIP-qPCR results indicated that AIPP3, AIPP2, and PAIPP3 were enriched in the selected target genes, particularly in the regions close to TSSs (Fig. 5a), but they were low in the adjacent high H3K4me3/low H3K27me3 genes. To confirm this point, an AIPP3 ChIP-seq was performed. As shown in Fig. 5b, AIPP3 specifically binds to the selected target genes. At the genome-wide level, AIPP3-binding peaks were enriched in the DEGs that were upregulated in the *aipp3-1* mutant (Fig. 5c, d). Moreover, a pattern of high H3K27me3 and low H3K4me3 was clearly observed in AIPP3-bound genes (Fig. 5e). Next, we compared the genome-wide occupancy of AIPP3 with a published global analysis of the H3K27me3-marked regions[1]. As shown in Fig. 5f, 455 loci, or approximately half of the AIPP3-enriched loci, significantly overlap with H3K27me3-enriched loci, indicating that a substantial part of the H3K27me3 loci were targeted by the AIPP3 complex. In *Arabidopsis*, EMBRYONIC FLOWER 1 (EMF1) is a plant-specific PRC1 component that is essential for conferring H3K27me3-dependent silencing at thousands of loci[29]. Recently, EMF1 has been shown to interact with two H3K27me3 readers to form the BAH–EMF1 complex and implement silencing[29]. We compared the AIPP3 loci with the published EMF1-bound loci and found that ~59% of AIPP3-enriched loci were also occupied by EMF1 (Fig. 5f). While, no EMF1 peptides were found in the BPC complex co-purified proteins (Supplementary Data 1). One possible explanation is that both EMF1 and AIPP3 associate with H3K27me3 mark independently, although the possibility of indirect association between these two reader proteins cannot be excluded.

To decipher whether H3K27me3 binding is indispensable for flowering time control and transcription repression, the wild-type and mutated AIPP3 genomic DNA, in which the crucial Tyr149, Trp170, and Tyr172 residues required for H3K27me3 binding were mutated into alanine was introduced into the *aipp3-1* mutant under the direction of the native promoter to generate *AIPP3*, *W170A*, and *Y149A/W170A/Y172A* transgenic *Arabidopsis*. The early flowering of the *aipp3-1* mutation was rescued by the wild-type *AIPP3* transgene, but not by the *W170A* or *Y149A/W170A/Y172A* transgenes (Fig. 6a, b). Interestingly, compared to the comparable accumulation levels of AIPP3 and W170A

proteins in transgenic plants, Y149A/W170A/Y172A protein level was much lower (Fig. 6c), indicating that Y149A/W170A/Y172A mutation alters the stability of AIPP3 protein. Therefore, we used *AIPP3* and *W170A* transgenes in the following experiment. RT-qPCR results indicated that the W170A mutation could not recover the repressive state of the selected target genes (Fig. 6d), suggesting that the H3K27me3-binding activity is essential for the repression of flowering and gene expression. To test whether W170A mutation has an impact on AIPP3 binding at target genes, ChIP-qPCR assay was performed in Col-0, *AIPP3*, and *W170A* transgenic plants. As shown in Fig. 6e, compared to the significant enrichment at selected target genes in AIPP3 transgene, AIPP3 binding was disrupted by the W170A mutation. Considering the fact that W170 is essential for H3K27me3-binding activity (Fig. 3h), this result strengthens our conclusion that H3K27me3-binding activity is indispensable for AIPP3-mediated flowering time control and transcriptional repression. Interestingly, the mutations did not affect the AIPP3 interactions with AIPP2 and PAIPP2 (Supplementary Fig. 11), indicating that the BPC complex is not dissociated by the disabling of H3K27me3 binding.

**BAH–PHD–CPL2 couples the recognition of H3K27me3 and the repression of Pol II release**. It has been well documented that differential phosphorylation on the Ser2 (Ser2P) and Ser5 (Ser5P) of the Pol II CTD plays essential roles in the switches between distinct transcriptional stages[64]. During transcription, Pol II is first assembled at the promoter region. After initiation, Pol II is phosphorylated at Ser5 and Ser2, and is then released from the proximal-promoter region to engage in productive elongation[64,65]. CPL2 has been shown to dephosphorylate the CTD-Ser5-PO4 of Pol II (ref. [66]). Therefore, it is reasonable to hypothesize that the BAH- and PHD-mediated recognition of H3K27me3 and unmodified H3K4 directly represses transcription through the CPL2-mediated dephosphorylation of Pol II. To confirm this hypothesis, we first determined the states of different forms of Pol II in the target genes. Recently, Zhu et al. revealed Pol II dynamics with single-nucleotide resolution in *Arabidopsis*, using native elongating transcript sequencing (NET-seq)[67]. Surprisingly, only a sharp peak of unphosphorylated Pol II signals was observed at the TSS region of the selected target genes, and both the Ser5P and Ser2P signals were quite low or even undetectable throughout the promoter-proximal regions and gene bodies (Fig. 7a). By contrast, in the H3K4me3-enriched active expressing genes that were independent from the regulation of the BPC complex (Fig. 4f), the maximum signals for

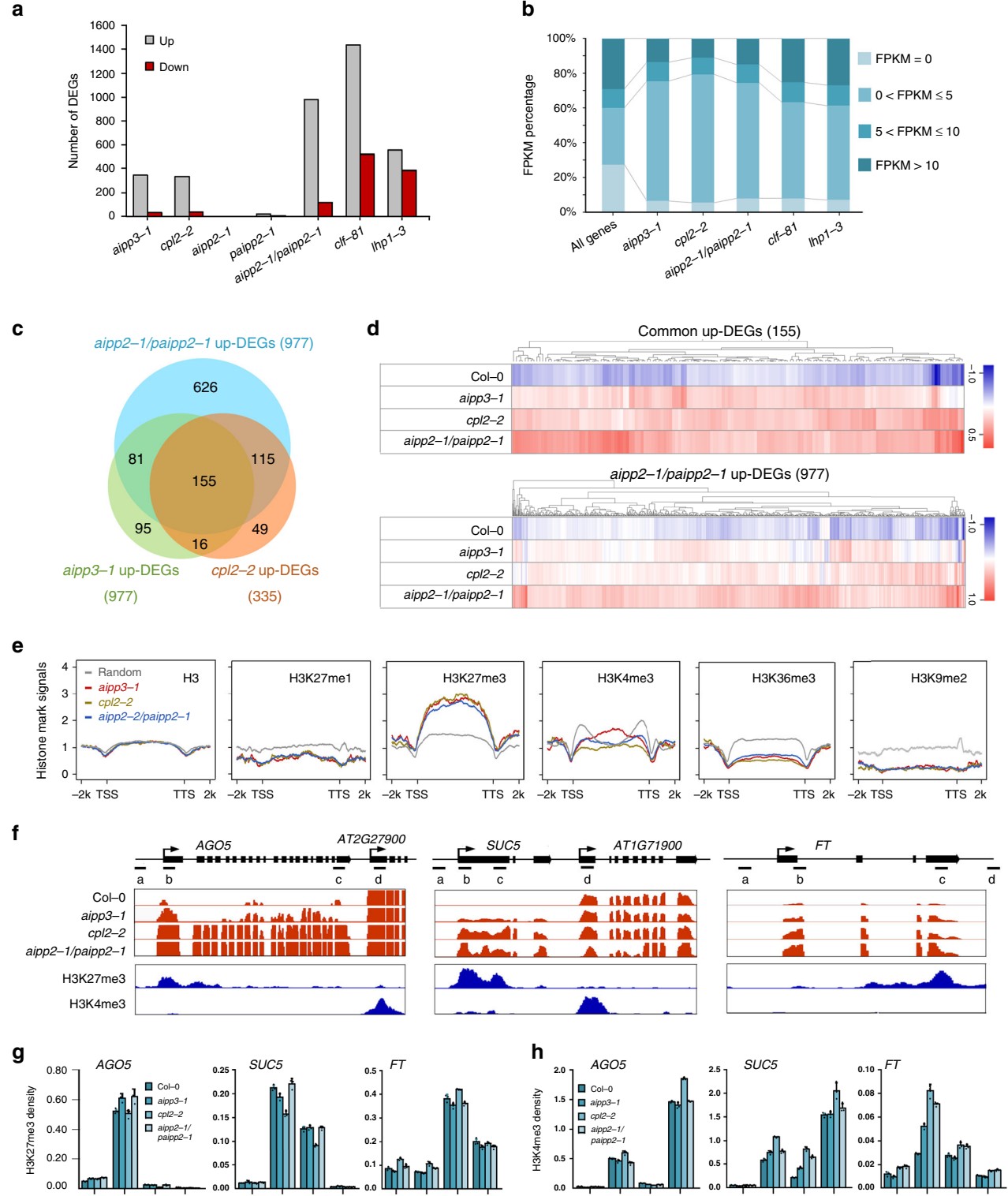

**Fig. 4 BAH–PHD–CPL2 complex represses the expression of H3K27me3-enriched genes. a** The numbers of up- and down-DEGs identified in the selected mutants. A twofold cutoff was used for the DEG criterion. **b** The BAH–PHD–CPL2 complex primarily represses low-expression genes. The gene expression levels of up-DEGs were classified into four groups according to their FPKM values in Col-0 mRNA-seq. All the annotated genes serve as controls. **c** A Venn diagram showing the overlap of up-DEGs between the selected mutants. **d** A heatmap showing the expression analysis in selected mutants. The genes in the upper and lower panels represent the common 155 up-DEGs and up-DEGs of *aipp2-1paipp2-1*, respectively. **e** The distribution of different histone marks in the respective up-DEGs of selected mutants. TTS transcription termination site. **f** Snapshots showing the expression of the selected target genes. One representative replicate of mRNA-seq and histone ChIP-seq are shown. The adjacent genes with high H3K4me3/low H3K27me3 were also shown as parallel controls. **g**, **h** ChIP-qPCR showing the H3K27me3 (**g**) and H3K4me3 (**h**) density at the selected target genes. The ChIP signals were normalized to histone H3. The data are means ± S.D. of three technical repeats. One representative result of three biological replicates is shown. The lowercase letters represent the ChIP-qPCR examined regions as shown in **f**.

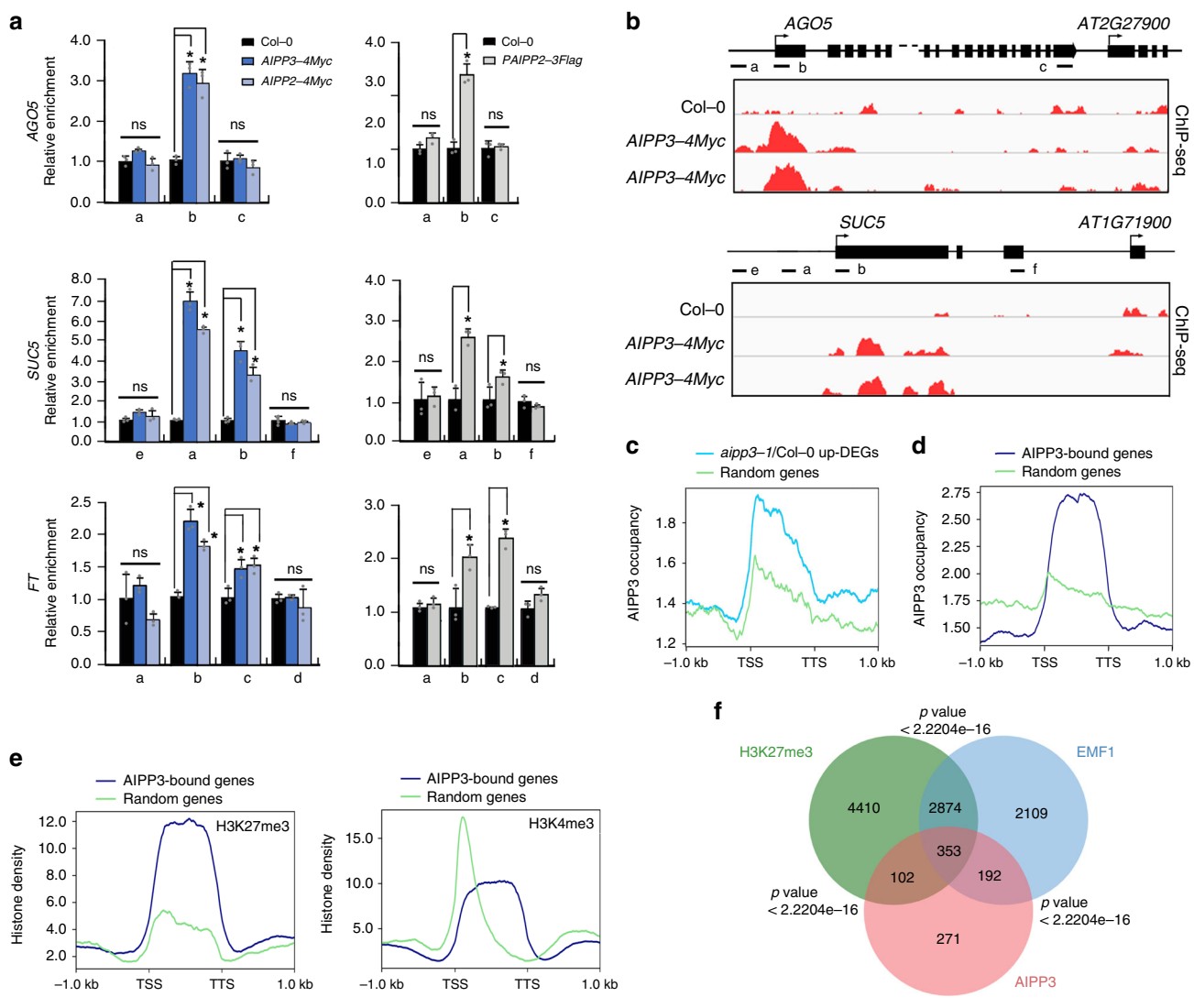

**Fig. 5 BAH–PHD–CPL2 complex binds to the H3K27me3-marked genes to repress the transcriptional initiation and elongation of Pol II. a** ChIP-qPCR showing the fold enrichments of AIPP3, AIPP2, and PAIPP2 on the selected genes. The occupancy was first normalized to the internal control *AtSN1*, and shown are relative fold enrichments compared to Col-0. The data are means ± S.D. of three biological repeats. Unpaired one-tailed *t* test was performed and *p value < 0.01. **b** Snapshots of AIPP3-4Myc ChIP-seq showing the distribution of the AIPP3 protein at the selected genes. Note: the ChIP-qPCR and ChIP-seq results of *AIPP3-4Myc* were from independent samples. **c, d** The diagrams showing AIPP3 occupancies on *aipp3-1* up-DEGs (**b**) and AIPP3-bound genes (**c**). **e** The diagrams showing the H3K27me3 (left) and H3K4me3 (right) density on AIPP3-bound genes. Random regions serve as negative controls. **f** A Venn diagram showing the overlap between AIPP3-bound genes, H3K27me3-enriched genes, and EMF1-bound genes.

unphosphorylated Pol II were detected at approximately +200 bp and sharply decreased to a very low level, and Ser5P and Ser2P-Pol II peaked in the promoter-proximal regions, but decreased to a mild level when entering the gene body regions (Fig. 7a). These results strongly support the idea that transcription initiation and subsequent elongation of target genes were repressed in the wild type by the BPC complex. We then checked what happens to the Pol II occupancy when the complex is absent. Compared to the wild type, higher accumulations of Ser5P-Pol II were observed at the selected target genes in *aipp3-1*, *cpl2-2*, and *aipp2-1/paipp2-1* mutants (Fig. 7b), and the Pol II signals maintained high levels toward the 3′ end of the selected genes. Similar to Ser5P-Pol II, the occupancy of total (unphosphorylated) and Ser2P-Pol II also displayed higher levels at selected target genes *AGO5*, *SUC5*, and *FT* in the *bpc* mutants in comparison with Col-0 (Supplementary Fig. 12). In contrast, the occupancies of all types of Pol II at *AGO5* and *SUC5* downstream nontarget genes were not significantly changed (Supplementary Fig. 12), demonstrating that *bpc*

mutations led to reactivation of Pol II initiation specifically at BPC complex target genes and the initiated Pol II successfully switched to an elongating state in the *bpc* mutants. These evidence, combined with the known knowledge that CPL2 is a Ser5P-Pol II phosphatase, prompts us to hypothesize that the BPC complex represses gene expression by connecting the BAH–PHD module-mediated recognition of H3K27me3/unmodified H3K4 and the CPL2-mediated dephosphorylation of Pol II CTD-Ser5-PO4.

To support the above hypothesis, flavopiridol (FLA) treatment assay was performed which can reduce the phosphorylation of Pol II CTD by inhibiting the activity of CDK kinases[67], and the nascent RNA levels of the selected target genes were measured by nuclear run-on assay. The results indicated that the nascent RNA levels of *AGO5* and *SUC5*, but not that of the downstream nontarget genes, were dramatically increased in the *bpc* mutants compared to Col-0 in mock condition. This evidence strongly supports our conclusion of transcriptional repression

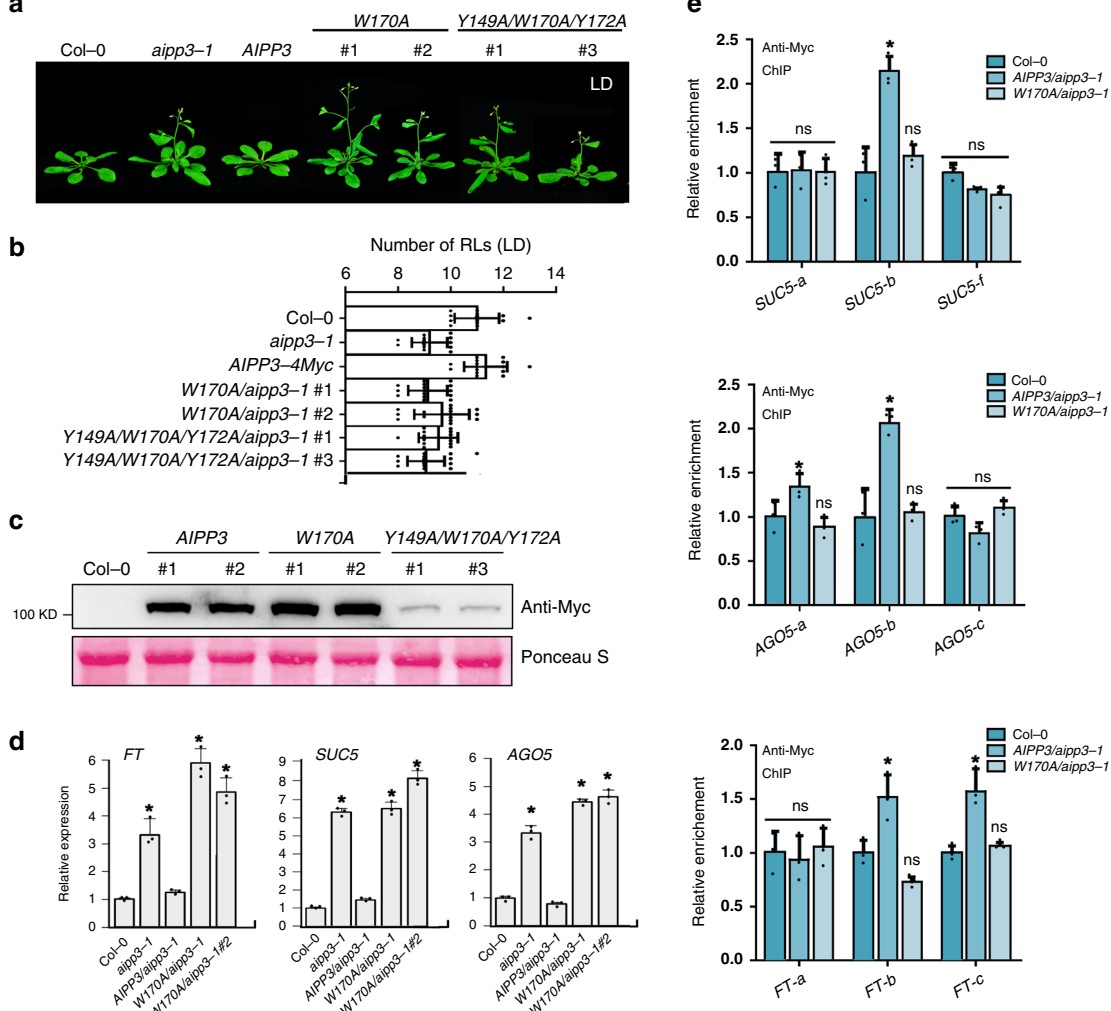

**Fig. 6 The H3K27me3-binding activity is essential for AIPP3-mediated repression of flowering and gene expression. a** The flowering phenotypes of the *aipp3-1* mutant for its complementary lines when transformed with the wild-type *AIPP3* genomic DNA, mutated *AIPP3* in which the key amino acids for H3K27me3-binding were mutated. For the mutated transgene, two randomly selected transgenes were used for the analysis. The plants were grown under LD. **b** The column showing the numbers of rosette leaves (LDs) of different *AIPP3* transgene plants upon flowering under the LD condition. Black horizontal lines represent the mean, and the error bars represent ±S.D. from the number of plants counted for each genotype. *n* = 15 per line. **c** Western blotting result showing the accumulation levels of AIPP3 proteins in different transgenes. Ponceaus staining serves as protein loading controls. **d** The relative mRNA levels of the selected target genes of the BAH–PHD–CPL2 complex in the wild-type, *aipp3-1*, and different *AIPP3* transgene plants. Their levels are presented relative to Col-0. The data are the means ± S.D. of three biological repeats. Unpaired one-tailed *t* test was performed and \**p* value < 0.01. **e** ChIP-qPCR resulting showing AIPP3 occupancy at the selected target genes in *AIPP3* and *W170A* transgenes. The data are the means ± S.D. of three biological repeats. Unpaired one-tailed *t* test was performed and \**p* value < 0.01. ns no significance.

implemented by the BPC complex. In contrast, the nascent RNA levels of both *AGO5*, *SUC5*, and the downstream nontarget genes were greatly reduced by FLA treatment, and no obvious changes were observed between different genotypes (Fig. 7c), indicating that inhibition of Pol II CTD phosphorylation dramatically repressed the transcription reactivation caused by the malfunctions of the BPC complex. Considering the likelihood of general effects of FLA treatment on transcriptional at a global level, to strengthen our hypothesis, the chromatin binding of CPL2 was compared in the presence or absence of AIPP3 and PHD proteins. To this end, *CPL2-GFP* transgene was crossed into *aipp3-1* and *aipp2-1/paipp2-1* mutants, and CPL2 ChIP-qPCR assay was performed in *CPL2-GFP*/Col-0, *CPL2-GFP/aipp3-1*, and *CPL2-GFP/aipp2-1/paipp2-1* plants. The result indicated that CPL2 has significant binding at selected target genes in wild-type background, whereas this binding was completely abolished

in the *aipp3-1* and *aipp2-1/paipp2-1* mutants (Fig. 7d). This result provides a link between chromatin marks and CPL2-mediated dephosphorylation of Pol II, in which the chromatin localization of CPL2 at BPC complex target genes largely depends on the recognition of H3K27me3/H3K4me0 marks by BAH–PHD proteins. Based on these evidences, we proposed a working model of the transcriptional repression conferred by the BPC complex (Fig. 8). In this model, the BAH–PHD bivalent histone reader recognizes H3K27me3, and unmodified H3K4 marks and recruits CPL2 to dephosphorylate the Ser5P of Pol II CTD, resulting in the inhibition of Pol II release from a transcriptional initiation state to elongation. When the BPC complex is absent, active H3K4me3 mark is deposited at BPC target genes. The Ser5 and Ser2 residues of Pol II CTD are sequentially phosphorylated, leading to transcriptional reactivation of BPC target genes.

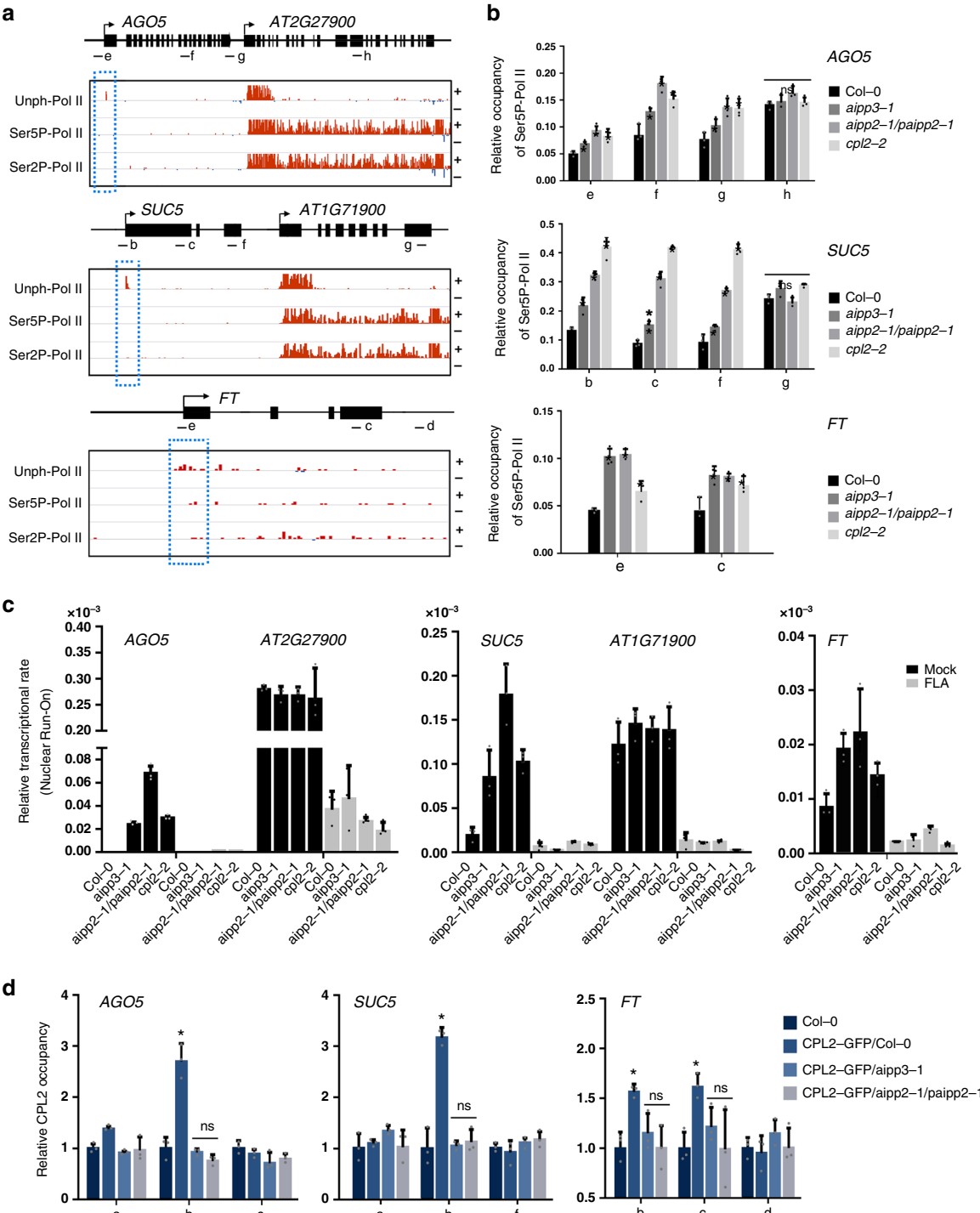

**Fig. 7 BPC complex directly connects H3K27me3 recognition with transcriptional repression. a** Snapshots showing the occupancies of different Pol II forms on selected target genes using a published Pol II NET-seq database. The red and blue lines indicate the Pol II signals in the plus and minus strands, respectively. The dashed red boxes indicate the occupancies of Pol II on the selected target genes. The black arrows indicate the transcriptional direction. **b** The ChIP-qPCR validation of the occupancy of Ser5P-Pol II on selected genes in different mutants. The lowercase letters the examined regions (the same below). The occupancy was normalized to the *ACT7*. The data are the means ± S.D. of three biological repeats. Unpaired one-tailed *t* test was performed and *p value < 0.01. ns no significance. **c** Nuclear run-on analysis showing the relative Pol II transcription rate at the selected target genes in Col-0 and *bpc* mutants with or without FLA treatment (mock). *AT2G27900* and *AT1G71900* genes serve as control genes. The relative transcription rate was normalized to 18 S rRNA. The data are the means ± S.D. of three biological repeats. **d** CPL2 ChIP-qPCR results showing the relative occupancy of CPL2 at selected target genes in the presence and absence of AIPP3 and AIPP2/PAIPP2. The occupancy was first normalized to AtSN1 and then normalized to Col-0. The data are the means ± S.D. of three biological repeats. Unpaired one-tailed *t* test was performed and *p value < 0.01.

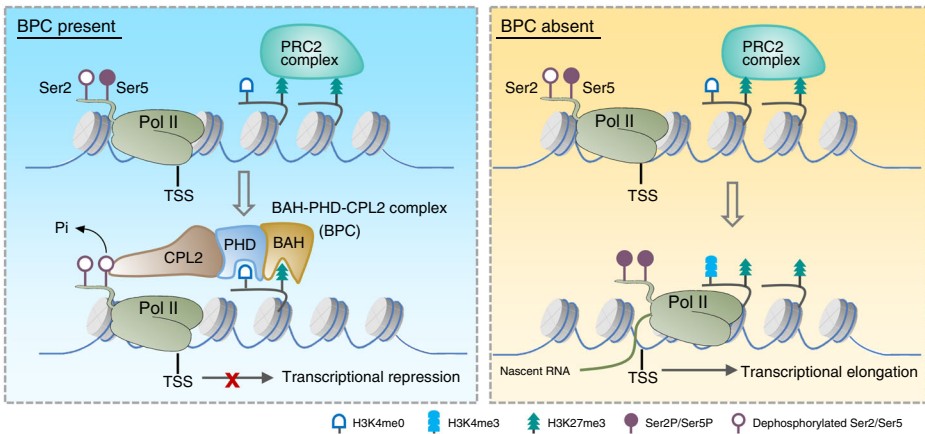

**Fig. 8 A working model of the BPC complex-mediated transcription repression.** When BPC is present, AIPP3-BAH, and AIPP2/PAIPP2-PHD motifs recognize H3K27me3, which is deposited by PRC2 complex, and unmodified H3K4 around the TSS, respectively. Then, the BAH–PHD histone reader module recruit CPL2 to dephosphorylate Pol II at the fifth Ser of CTD, thereby repressing the transcriptional initiation and subsequent elongation of Pol II. When BPC is absent, active H3K4me3 mark is deposited. Pol II CTD-Ser2 and -Ser5 residues can be phosphorylated sequentially, leading to release of Pol II from initiation to elongation state.

## Discussion

Recently, we structurally described several plant H3K27me3-reading BAH domain proteins, such as EBS, SHL, and AIPP3 in this research[27,28], which has encouraged us to characterize the H3K27me3-reading BAH domains. The recognitions of H3K27me3 by all these BAH domains depend on the aromatic cage to recognize the methyl-lysine, and a histidine and an aspartic acid residue to specifically interact with H3P30 for sequence specificity. Using the aromatic cage, and the specific His and Asp residues as a criterion, we identified a subfamily of BAH domain proteins that can potentially recognize the H3K27me3 mark, which are widely distributed in plants, fungi, and animals (Supplementary Fig. 13). Interestingly, the predicted H3K27me3-binding BAH domain-containing protein human BAHD1 was reported to be an H3K27me3 reader[30], which further supported our prediction. Therefore, we believe that the aromatic cage, and conserved His and Asp residues are the key features of H3K27me3-recognition BAH domains.

Different histone marks do not function in a totally independent manner. Instead, they engage in communications and cooperation with each other. The cooperation between different histone marks can be a combinatorial reading to enhance binding or be mutually exclusive to balance the binding between different marks, which occurs at both the single protein level and the multiple protein complex level[68]. Recently, we reported that the two flowering regulators EBS and SHL could dynamically recognize the antagonist histone marks H3K27me3 and H3K4me3 at the single protein level to regulate floral phase transitions[27,28]. Here, AIPP3 and AIPP2/PAIPP2 could recognize H3K27me3 and H3K4me0, respectively, and form a bivalent histone reader complex. This observation is consistent with our functional data showing that the BPC complex colocalizes with higher H3K27me3 and lower H3K4me3 marked chromatin regions. The BAH domain of AIPP3 is responsible for the colocalization of the complex with H3K27me3. The binding of unmodified H3K4 by the PHD finger of AIPP2/PAIPP2 may have two roles. First, the binding of unmodified H3K4 may prevent the binding toward methylated H3K4 to make sure the complex is targeted to the gene repressive H3K4me3 depletion region. Second, the binding of H3K4me0 together with the H3K27me3 binding by AIPP3 may combine to enhance the overall binding of the complex toward a certain chromatin region. Therefore, the

proper targeting of the BPC complex relies on the crosstalk between the H3K4me0 and H3K27me3 marks.

RNA Pol II-dependent transcription is a stepwise process involving the formation of the preinitiation complex (PIC), initiation, elongation, and polyadenylation/termination stages. Each of the stages is associated with a distinct pattern of CTD phosphorylation[8,65]. The transcriptional machinery is first recruited to the promoter regions. Once incorporated into the PIC, the mediator stimulates cyclin-dependent kinase to phosphorylate serine 5 of the CTD heptad repeat, and Ser5P helps in the release of Pol II from the PIC complex, thereby allowing Pol II to escape the promoter and the subsequent initiation of transcription. Ser5P is retained during the first several hundred nucleotides, in preparation for productive elongation[69].

Here, on the selected target genes *AGO5* and *SUC5*, unphosphorylated Pol II is restricted in the TSS region, whereas Ser5P and Ser2P were below detectable levels (Fig. 7a), indicating that Ser5P-dependent transcription initiation is inhibited, possibly by CPL2, in the presence of the BPC complex. Consistent with this notion, the density of the Ser5P-Pol II was significantly increased, and it peaked within the first several hundred nucleotides in the BPC complex mutants (Fig. 7b), indicating that the Ser5P-dependent transcription initiation was derepressed. In the canonical model of H3K27me3-mediated transcription repression, the recognition of H3K27me3 recruits the PcG proteins in the PRC1 complex to impose the monoubiquitination of H2A, which represses transcription through three possible mechanisms, as mentioned in the introduction[8]. Consistent with this model, a recent report showed that the mutations in H3K27me3 reader proteins LHP1, EBS, and SHL in *Arabidopsis* led to a reduction in H2Aub1 (ref. [29]). We found that the H2Aub1 levels were reduced in the *lhp1-3* mutant, but were not affected in the *aipp3-1/cpl2-2*, *aipp3-1/aipp2-1/paipp2-1*, and *aipp2-1/paipp2-1/cpl2-2* mutants (Supplementary Fig. 10a). This observation is consistent with our hypothesis that the BPC complex represses transcription mainly through the inhibition of Ser5P-dependent transcription initiation. While, we cannot rule out the possibility that BPC the complex has direct/indirect interaction with H2Aub1 at specific target genes. This study unveiled a direct connection between H3K27me3 recognition and transcription repression; the BAH–PHD module recognizes H3K27me3/H3K4me0 and recruits CPL2 to dephosphorylate Pol II CTD, resulting in the

failure of Pol II to enter into the initiation and elongation form. Although Pol II CTD phosphatases are conserved in eukaryotes, no evidence shows that histone modifiers manipulate CTD phosphatases, including CPL2 analogies in other systems, to affect transcription. Regarding histone modifiers, several studies have reported that histone PTM modifiers can affect transcription via modulating Pol II CTD phosphorylation state[70,71]. JMJD3, an H3K27me3 demethylase in human, has been shown to directly interact with CTD-Ser2P to affect gene expression[72]. Knocking down JMJD3 or JHDM1D, a H3K27me1/2 demethylase, reduces the enrichment of Pol II CTD-Ser2P at specific genes in human promyelocytic leukemia cells[73]. In addition to histone methylation, other histone PTMs including histone ubiquitylation and phosphorylation, are also associated with Pol II CTD phosphorylation-dependent transcriptional elongation[71]. For example, knocking down histone ubiquitylation modifiers has been shown to affect CTD-Ser2 phosphorylation[71]. The phosphorylation of histone H3 on S10 and S28 has been reported to be associated with phosphorylated Pol II during transcriptional activation in humans and *Drosophila*[74,75]. These studies support a notion that modulation of Pol II CTD phosphorylation represents an important regulatory mechanism adopted by chromatin regulators to regulate gene expression. Our finding that the BPC complex reading histone information and conferring transcriptional repression through CPL2 phosphatase-mediated modulation of Pol II CTD phosphorylation state provides a direct evidence to support this notion. Considering that CPL2 is a plant-specific Pol II phosphatase that bears a unique RBM domain, and that this domain is required for its interaction with PHD proteins (Fig. 1c), the BAH–PHD–CPL2 pathway may represent a newly evolved silencing pathway in plants. Our findings suggest a greater complexity and diversity of H3K27me3-mediated transcriptional regulation. In addition, it is well-known that H3K27me3-mediated silencing mechanisms participate in multiple biological processes in both plants and animals, particularly development and stress-responsive genes. The obvious developmental defects (Fig. 2a) and reactivation of many development- and stress response-related genes (Supplementary Fig. 9) in *bpc* mutants imply that the BPC complex may play more important roles in plant development and stress responses in addition to flowering time control.

## Methods

**Plant materials and growth conditions**. All the plant seeds were sown and grown on 1/2 MS medium containing 1% sucrose. The seedlings were grown under a LD (16 h light/8 h dark) or SD (8 h light/16 h dark) photoperiod at 23 °C. The T-DNA insertion mutants *aipp2-1*, *aipp3-1*, and *cpl2-2* were described in our previous study[41]. The *paipp2-1* mutant was generated by CRISPR/Cas9-mediated mutagenesis in a Col-0 background (Supplementary Fig. 4). *clf-81* and *lhp1-3* have been described previously[53,54]. For the epitope-tagged transgenic expression of the *AIPP2*, *AIPP3*, *PAIPP2*, and *CPL2* genes, the wild-type and mutated genomic DNA driven by their native promoters were cloned into binary vectors with different tags and then transformed into the corresponding mutants, using the flowering dip method. T3 generation transgenic plants were used for analysis.

FLA treatment assay was performed as previous report[67]. In brief, 2-week-old seedlings were collected and incubated with 200 μM FLA or mock solution (DMSO) overnight. The treated seedlings were subjected to total RNA extraction.

**RT-qPCR and RNA-seq analysis**. Total RNA was extracted from 2-week-old seedlings using Trizol reagent (Thermo), and cDNAs were synthesized using HiScript II Reverse Transcriptase (Vazyme). qPCR was performed using a CFX96 Touch Deep Well Real-Time PCR Detection System (Bio-Rad). Three biological replicates were created. The primers used in this study are listed in Supplementary Table 2. For the RNA-seq analysis, total RNAs were extracted from 12-day-old seedlings grown during LDs using a RNeasy Plant Mini kit (Qiagen). Following RNA purification, reverse transcription and library construction, the libraries were quantified by TBS380, and a paired-end RNA sequencing library was performed with Illumina NovaSeq 6000 (2 × 150 bp read length). The raw paired-end reads were trimmed and subject to quality control with SeqPrep (https://github.com/

jstjohn/SeqPrep) and Sickle (https://github.com/najoshi/sickle), using the default parameters.

**IP and MS analysis**. IP and MS analyses were performed as previously described[76]. In brief, the total proteins were extracted from the inflorescence tissues with IP buffer (50 mM Tris-HCl, pH 7.6, 150 mM NaCl, 5 mM MgCl₂, 10% glycerol, 0.1% NP-40, 0.5 mM DTT, and protease inhibitor cocktail), and then precipitated with anti-Flag (Sigma-Aldrich) or anti-Myc (Millipore) antibodies for 2 h at 4 °C. After five times washing, the precipitated protein mixtures were subjected to MS analysis.

**Protein interaction analysis and gel filtration assays**. For the Y2H assays, the full-length and truncated coding sequences of *AIPP2*, *AIPP3*, *CPL2*, and *PAIPP2* were cloned into the pGADT7 and pGBKT7 vectors to generate AD and BD constructs. After the transformation, the yeast cultures were spotted onto SD plates lacking Trp and Leu (-LW) or lacking Trp, Leu, and His (-LWH) and incubated at 30 °C for 3 d.

For gel filtration assay, the total proteins were extracted from 4 g of seedling tissues expressing *AIPP3-FLAG*, *AIPP2-4MYC*, *PAIPP2-3FLAG*, and *CPL2-4MYC* with IP buffer and loaded on to a Superdex 200 10/300GL column (GE Healthcare). The eluted fractions were collected in 96-well plates and the target proteins were detected by standard western blotting.

**Protein expression and purification**. The sequence containing the AIPP3-BAH domain (residues 112–279) was constructed into a self-modified pMal-p2X vector to fuse a hexahistidine tag plus a maltose-binding protein (MBP) tag to the N-terminus of the target protein. The plasmids were transformed into *Escherichia coli* strain BL21 (DE3) RIL (Stratagene). Expression was induced at 16 °C overnight with 0.2 mM of IPTG. The recombinant proteins were purified with a prepackaged HisTrap FF column (GE Healthcare). The His-MBP tags were cleaved by TEV protease overnight and removed by flowing through a HisTrap FF column (GE Healthcare) again. The target protein was further purified using a Heparin column (GE Healthcare) and a Superdex G200 column (GE Healthcare). All the AIPP3-BAH mutations were generated by standard PCR-based mutagenesis procedure. The mutations of AIPP3-BAH and truncated AIPP2/PAIPP2 fragments were purified using the same protocols, as those used for the wild-type AIPP3-BAH. For the GST-AIPP3-BAH proteins used in histone peptide pull-down assays, the wild-type and mutated AIPP3-BAH proteins were purified with glutathione-Sepharose (GE Healthcare) and eluted with elution buffer (50 mM Tris-HCl pH 8.0, and 10 mM reduced glutathione). The peptides were purchased from GL Biochem or EpiCypher.

**Histone peptide pull-down**. For histone peptide pull-down assay, 1.5 μg of biotinylated histone peptides were incubated with streptavidin beads (NEB) in binding buffer (50 mM Tris-HCl 8.0, 300 mM NaCl, and 0.1% NP-40) for 1 h at 4 °C, and then washed with binding buffer. A 1.5 μg quantity of AIPP3-BAH proteins was incubated with a peptide–bead mixture in 0.5 ml of binding buffer for 3 h at 4 °C. and then washed with binding buffer five times. The protein–bead mixtures were subjected to immunoblotting using anti-GST antibody (Abmart, #12G8, 1:2000 dilution).

**Chromatin immunoprecipitation assays and ChIP-seq analysis**. ChIP assays were performed according to a reported procedure[77]. In brief, 3 g of seedlings was harvested and fixed with 1% formaldehyde in the cross-linking buffer (0.4 M sucrose, 10 mM Tris-HCl pH 8, 1 mM PMSF, and 1 mM EDTA). After nuclei isolation with isolation buffer (0.25 M sucrose, 15 mM PIPES pH 6.8, 5 mM MgCl₂, 60 mM KCl, 15 mM NaCl, 1 mM CaCl₂, 0.9% Triton X-100, and protease inhibitor cocktail) and the following centrifugation, nuclei were resuspended in 500 μl of lysis buffer (50 mM HEPES pH 7.5, 150 mM NaCl, 1 mM EDTA, 1 mM PMSF, 1% SDS, 0.1% Na deoxycholate, 1% Triton X-100, and protease inhibitor cocktail) and sonicated with a Bioruptor (Diagenode). The nuclei lysate was precipitated with anti-Flag (Sigma-Aldrich, #F1804, 1:100 dilution), anti-Myc (Millipore, #05-724, 1:100 dilution), anti-H3 (Abcam, #ab1791, 1:100 dilution), anti-H3K27me3 (Millipore, #07-449, 1:100 dilution), Ser2P-Pol II (Abcam, #ab5095, 1:100 dilution), Ser5P-Pol II (Abcam, #ab5131, 1:100 dilution), unphosphorylated Pol II (Abcam, #ab817, 1:100 dilution), and GFP (Abcam, #ab290, 1:100 dilution) antibodies overnight and incubated with Dynabeads (Thermo) for 2 h. The precipitated protein–DNA mixtures were washed and eluted with elution buffer (0.5% SDS and 0.1 M NaHCO3) at room temperature. The DNA was recovered after reverse cross-linking and proteinase K treatment.

For ChIP-seq analysis, clean reads were mapped to the *Arabidopsis thaliana* genome (TAIR10) by Bowtie2 (version 2.2.8) with default parameters[78]. Enriched peaks were identified by MACS (version 1.4) with default parameters. We defined the region of a target gene as the range from 1 kb upstream of TSS to TTS. The target genes of each peak were annotated by annotatePeak function in ChIPseeker package. The visualization of the average read coverage over gene body and additional 1 kb upstream and downstream of the TSS and TES was performed by deepTools (version 2.4.1)[79].

**Crystallization, data collection. and structure determination**. Crystallization screening was performed using the sitting drop vapor diffusion method at 4 °C. The sample was concentrated and mixed with peptide at a molar ratio of 1:4. All the crystals emerged in a solution of 0.1 M HEPES, pH 7.0, and 2.4 M ammonium sulfate. The crystals were cryo-protected in the reservoir solution supplemented with 20% glycerol and flash-cooled in liquid nitrogen for X-ray diffraction. All the diffraction data were collected at beamline BL19U1 of the National Center for Protein Sciences Shanghai (NCPSS) at the Shanghai Synchrotron Radiation Facility (SSRF). The data were processed with the HKL3000 program[80]. The structure was determined by molecular replacement method as implemented in the Phenix program[81], using the ZMET2 BAH domain (PDB ID: 4FT2) as the searching model[20]. Model building and structure refinement were performed using the Coot and Phenix programs, respectively[81,82]. The statistics of data collection and structure refinement are listed in Supplementary Table 1.

**ITC calorimetry**. All the binding experiments were performed on a Microcal PEAQ-ITC instrument (Malvern) at 25 °C. The proteins were dialyzed against a buffer consisting of 50 mM NaCl and 20 mM Tris-HCl, pH 7.5 overnight at 4 °C. The lyophilized peptides were dissolved into the dialysis buffer. The titration was performed using the standard protocol and the data were processed using the Origin 7.0 program.

**Nuclear run-on assay**. Nuclear run-on assay was performed as previous report with minor changes[83]. In brief, after nuclei isolation with buffer (0.25 M sucrose, 15 mM PIPES pH 6.8, 5 mM $MgCl_2$, 60 mM KCl, 15 mM NaCl, 1 mM $CaCl_2$, 0.9% Triton X-100, and protease inhibitor cocktail), the nuclear pellet was resuspended with nuclei storage buffer (50 mM Tris-HCl pH 8, 0.1 mM EDTA, 5 mM $MgCl_2$ and 40% glycerol) and mixed with transcription buffer (10 mM Tris-HCl, pH 8, 2.5 mM $MgCl_2$, 150 mM KCl, 5 mM DTT, 80 units RNase inhibitor, 0.5 mM BrUTP, 1 mM ATP, 1 mM GTP, 1 mM CTP, 0.5 mM UTP, and 0.2% sarkosyl). After incubation at 30 °C for 15 min, the run-on reaction was stopped by adding Trizol reagent (Thermo). The nuclear RNAs were extracted and treated with DNase. The DNA-free RNAs were then inoculated with 2 μg anti-BrdU antibody (abcam) at room temperature for 30 min and precipitated by Dynabeads (Thermo). After two times washing, RNAs were extracted using Trizol reagent (Thermo). cDNAs were synthesized using SuperScript IV Reverse Transcriptase (Thermo) and subjected to qPCR analysis.

**GUS staining assay**. The plant materials were stained in staining solution overnight at 37 °C and then washed with 70% ethanol at room temperature. The plants were cleared after being washed and were observed under a ZEISS Stemi 305 microscope.

**Histone immunoblotting**. The nuclei were extracted from 2-week-old seedlings with isolation buffer (0.25 M sucrose, 15 mM PIPES pH 6.8, 5 mM $MgCl_2$, 60 mM KCl, 15 mM NaCl, 1 mM $CaCl_2$, 0.9% Triton X-100, and protease inhibitor cocktail) and resuspended with lysis buffer (50 mM HEPES pH 7.5, 150 mM NaCl, 1 mM EDTA, 1 mM PMSF, 1% SDS, 0.1% Na deoxycholate, 1% Triton X-100, and protease inhibitor cocktail) to release the total chromatin. The supernatant was boiled with SDS loading buffer and subjected to immunoblotting with anti-H3 (Abcam, #ab1791, 1:1000 dilution), anti-H3K27me1 (Millipore, #07-448, 1:1000 dilution), H3K27me2 (Millipore, #07-452, 1:1000 dilution), H3K27me3 (Millipore, #07-449, 1:1000 dilution), anti-H3K4me1 (Millipore, #07-436, 1:1000 dilution), anti-H3K4me2 (Millipore, #07-030, 1:1000 dilution), anti-H3K4me3 (Millipore, #07-473, 1:1000 dilution), and anti-H2AKub (Cell Signaling, #8240, 1:1000 dilution) antibodies.

**Split luciferase assay**. For split luciferase assay, the constructs fused with split luciferase were co-infiltrated into tobacco (*Nicotiana benthamiana*) leaves. The luciferase activity was determined at 2 days after infiltration using CCD camera (Princeton instruments).

**Reporting summary**. Further information on research design is available in the Nature Research Reporting Summary linked to this article.

## Data availability

Coordinates and structure factors have been deposited in the RCSB Protein Data Bank with the accession code: 7CCE. The ChIP-seq and mRNA-seq data have been deposited in the GEO with the accession codes GSE147981 and GSE157196, respectively. All other data are available from the corresponding authors on request. Source data are provided with this paper.

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

## Acknowledgements

We thank Dr. Yuehui He for the seeds of *clf-81*, *lhp1-3*, and *flc-3* mutants. We thank staff at beamline BL19U1 of the National Center for Protein Sciences Shanghai (NCPSS) at the Shanghai Synchrotron Radiation Facility (SSRF) for data collection. C.-G.D. was supported by the Strategic Priority Research Program of the Chinese Academy of Sciences (XDB27040203) and by the National Natural Science Foundation of China (31570155). J.-K.Z. was supported by the Strategic Priority Research Program of the Chinese Academy of Sciences(XDB2704). J.D. was supported by the National Natural Science Foundation of China (31770782), Shenzhen Science and Technology Program (JCYJ20200109110403829 and KQTD20190929173906742), SUSTech (G02226301), and the Key Laboratory of Molecular Design for Plant Cell Factory of Guangdong Higher Education Institutes (2019KSYS006).

## Author contributions

C.-G.D., J.D., and J.-K.Z. designed this study. Y.-Z.Z., J.Y., L.Z., Y.W., G.Z., and S.-S.X. conducted the experiments. Y.Z.Z., J.Y., L.Z., J.J., J.D., J.-K.Z., and C.-G.D. analyzed the

data. C.C. and P.L. performed bioinformatic analysis. J.D., J.-K.Z., and C.-G.D. wrote the paper.

## Competing interests

The authors declare no competing interests.
