## [Peer Review File · Nature Communications]

REVIEWER COMMENTS

Reviewer #1 (Remarks to the Author):

In this study, Zhang et al identify the BAH domain of AIPP3 and the PHD domain of AIPP2/PAIPP2 as the specific reader module of H3K27me3 and unmodified H3K4 in Arabidopsis, respectively; AIPP3 and AIPP2/PAIPP2 further associate with a plant-specific Pol2 carboxyl terminal domain (CTD) phosphatase CPL2 for targeted gene repression by altering Pol2 phosphorylation state. The authors verify that AIPP3 associates with AIPP2 and CPL2 to form a ternary complex (BAH-PHD-CPL2) in Arabidopsis, and genes in this complex is shown to act in the same pathway in flowering time control. They verify AIPP3-BAH domain as a H3K27me3 reader module by ITC and determine the crystal structure of BAH in complex with an H3K27me3 peptide. Likewise, PAIPP2-PHD is confirmed as a reader of unmodified H3K4; key residues involved in interaction are identified by structure modelling. They demonstrate that BAH-PHD-CPL2 represses the expression of H3K27me3-enriched genes, and that H3K27me3 binding is indispensable to transcription repression and flowering time control by structure-based mutagenesis in Arabidopsis. NET-seq and flavopiridol assay are performed to provide evidence that transcription of targeted gene transcription is repressed by BAH-PHD-CPL2. These data support a model that BAH-PHD-CPL2 bridges the cross-talk between H3K27me3 and Pol2 phosphorylation, which plays an important role in flowering time control in Arabidopsis.

This is a well-presented manuscript displaying a new mechanism of H3K27me3 mediated transcription regulation, which will be interesting to a broad audience. I recommend the acceptance of the manuscript for publication. Here are some modest questions and suggestions:

1. The authors model the AIPP2 and PAIPP2 PHD fingers based on the structure of ATXR5 PHD finger to analyze the interaction between unmodified H3K4 and AIPP2/PAIPP2 PHD fingers, and identify several conserved residues which may involve in the recognition. I wonder if the author can provide further evidence to support their modelling result, such as ITC using the mutants.
2. Line 413-416: I would suggest the author providing a ChIP-qPCR result of AIPP3, W170A, Y149A/W170/Y172 in aipp3-1 mutant strain to strengthen their conclusion that H3K27me3 by AIPP3 is indispensable for flowering time control and transcription repression. In Line 413, I guess "Y149A" is a typo and should be "W170A"?
3. Line 448: I wonder what is the level of Ser2P Pol2 in aipp3-1, cpl2-2 and aipp2-1/paipp2-1 mutants. It would be more informative to compare the fluctuation of Ser2P and Ser5P in the mutants since CPL2 was reported as a Pol CTD Ser5 phosphatase.

Other comments:

1. Line 327: "155 genes were commonly up-regulated by the BAH-PHD-CPL2 complex". I guess it should be "in the bah-phd-cpl2 mutant"?
2. Fig5a left panel: There is a shift of genome site label "e, a, b, f".
3. Line 384 and Fig 5e: "... AIPP3-enriched loci, significantly overlap ..." p-value should be provided as the authors mention "significantly overlap".

Reviewer #2 (Remarks to the Author):

This is an interesting study that progresses our understanding of mechanisms of how chromatin

modifications influence transcription. The authors characterize an H3K27me3 reader, first using a peptide pull-down assay and then ITC. They continue by defining the BAH structure by co-crystallizing the BAH domain with H3K27me3. They corroborate and validate the histone recognition with a mutation analysis to show how the binding is affected after mutation of the aromatic cavity that provides an optimal accommodation of K27me3.

They also analyse the PHD fingers PAIPP2 and AIPP2-PHD. Alignments and structural predictions for the PHDs indicate a possible interaction with unmodified H3K4 and this was confirmed for PAIPP2-PHD by ITC.

Specific comments:

1) Lines 218-220: the ITC graph they refer to in Fig. 3 c shows Kd values that match the sequential increase, however the curves in this graph do not correspond to the Kds – is there a mixup in the colouring of dark blue and light blue in the legend?

2) lines 388-393 and lines 555-559: in the results section the authors make a connection of their complex to PRC1, but they only show that many AIPP3-enriched loci overlap with EMF1-bound loci. Since both associate with H3K27me3 this could just be a correlation. If EMF1 is found in the IP-MS dataset for AIPP3, this would make the claims more convincing. The double negative “not a histone ubiquitination-independent mechanism” in line 559 needs rephrasing. Showing global levels of H2Aub is not enough to support a broad statement on dependency of H2Aub (eg. they might only see differences when they look at their specific target genes), so this section of the text needs to be carefully edited.

3) lines 409-421: in connection to H3K27me3 binding of AIPP3, they introduce mutated AIPP3 transgenes (W170A, Y149A/W170A, Y172A) into the aipp3-1 mutant and say that the mutated transgenes do not complement the flowering time phenotype and the repressive state of the selected target genes. However, this would also be true for transgenes that do not produce a stable protein independent of any binding change. A western showing that protein levels of AIPP3 are restored in mutated AIPP3 transgenic lines is essential and ideally also a ChIP experiment showing that chromatin binding of mutated AIPP3 is lost at selected target genes as a more direct evidence for in vivo H3K27me3 binding.

4) Lines 452-466, on coupling with Pol II release: to support the coupling hypothesis they rely on FLA treatments based on the previous report that CPL2 dephosphorylates CTD Ser5. However, this experiment cannot be used to directly link BAH-PHD-CLP2 complex activity to phosphorylation of Pol II, because of the likelihood of general, indirect effects on transcription at a global level.

Minor issues:

-the introduction in particular needs editing for english

-there is no reference to datasets in the text - full datasets for IP-MS (Fig. 1), RNAseq (Fig. 4) and ChIP-seq of AIPP3 (Fig. 5) should be provided.

- line 161-163, not all phenotypic developmental defects described for aipp3, cpl2 and aipp2/paipp2 mutants are visible from the images shown (Fig. 2a)

-line 247, the reference to Supplementary Fig. 3b should be Suppl. Fig. 7

-For Fig. 4b, does the calculated FPKM percentage only refer to up-DEGs in the mutants? This is not clear from the figure legend.

- In the supplementary Fig.8a AtPHD2-PHD should be changed to AtPAIPP2-PHD.

Reviewer #3 (Remarks to the Author):

In this manuscript, Zhang and colleagues report the functional characterisation of the Arabidopsis BAH-PHD-CPL2 complex (BPC complex), defining its role in transcriptional gene silencing. BPC is composed by the PHD-domain proteins AIPP2/PAIPP2, the BAH-domain protein AIPP3 and the Pol CTD Ser5 phosphate CPL2. Each complex component has a differential activity: 1) recognition of the PRC2 repressive histone modification H3K37me3 by BAH domains, 2) recognition of H3K4me3 by PHD domains, 3) Pol II CTD dephosphorylation by CPL2. By conducting different protein-interaction analyses (including mass-spectrometry, Y2H, luciferase split assay, gel filtration

followed by immunoblot analysis), the authors very robustly showed the interaction of AIPP2 and PAIPP2 with AIPP3 and CPL2 proteins. Based on the data, this work proposes the existence of two types of BPC complexes, AIPP2/AIPP3/CPL2 and PAIPP2/AIPP3/CPL2, with AIPP2 and PAIPP2 having redundant functions. The authors confirmed the capacity of AIPP3-BAH domain to recognize H3K27me3 using different approaches, including structural studies. Additionally, this work demonstrates the capacity of AIPP2- and PAIPP2-PHD domains to bind to unmethylated H3K4. BPC complexes seem to play a very important role in Arabidopsis development. The *aipp3* and *cpl2* single mutants, as well as the *aipp2/paipp2* double mutants, are early flowering both in long and short days. The early flowering phenotype correlates with increased levels of FT mRNA in *aipp3*, *cpl2* and *aipp2/paipp2*, and was rescued by the *ft-10* mutation. In addition, combinations of triple mutant analyses showed similar phenotypes as those of the single mutants, demonstrating that the BPC complex components belong to the same genetic pathway. RNA-seq and ChIP-qPCR assays revealed a connection of BPC with gene repression, as well as a link to PRC2-mediated silencing. Furthermore, the authors demonstrated that AIPP3 target genes are enriched with the H3K27me3 repressive mark, whereas H3K4me3 levels are reduced at those genes. Interestingly, the absence of BPC leads to accumulation of the phosphorylated version of RNA Pol II that is involved in productive transcription elongation (Ser5P Pol II) over two selected BPC target genes. Therefore, this work documents that the combined activities of the three BPC complex components result in coupling of H3K27me3 recognition with direct regulatory modifications of RNA Pol II, eventually leading to transcriptional repression. This manuscript proposing a direct connection of H3K27me3 to inhibiting RNA Pol II elongation represents an interesting advance to the PRC2-mediated gene silencing mechanism in plants. I have some comments that should be addressed to improve this manuscript.

Major points:

- 1- Data presented in this work suggest that FT is an important target of BPC complex, explaining the early flowering phenotype of the *aipp3*, *cpl2* and *aipp2/paipp2* mutants. Is FT a direct target of BPC? Including FT locus in the ChIP-seq analysis shown in Figure 5A would help to clarify this point. Similarly, it would be good to include FT locus in Figure 6B-C.
- 2- As the authors claim that BPC is driving transcriptional repression, it would be more suitable to measure the levels of nascent (unspliced) RNAs in Figure 6C.
- 3- To fully support the idea of Ser5 CTD dephosphorylation by the CPL2 module of BPC, I would suggest that the authors test enrichment of total Pol II (unphosphorylated) and Ser2 by ChIP-qPCR, comparing Col0 vs *bpc* mutants. In addition, in the analysis of Figure 6B, including primers over the downstream genes (AT2G27900 and AT1G71900) would help to define whether the differences shown are significant.
- 4- I think that the general message and model proposed (Figure 6D) in this work could be further improved. Currently, a general repression mechanism is proposed, linking PRC2 repressive activity to preventing RNA Pol II productive elongation by the action of CPL2. In my view, the function of BPC is preventing Pol II elongation in loci that were already silenced by PRC2 (were already covered by H3K27me3). In other words, a surveillance system to prevent gene reactivation. This could be supported by the fact that no changes in H3K27me3 were observed in the selected genes in the different BPC mutants analysed (Figure 4G), and yet mRNA levels increase (Figure 4F). In this regard, testing H3K27me3 enrichment over more BPC targets in the different mutants would strengthen this work. Also, and similarly to that stated above, it would be good to provide data for nascent transcripts.

Minor comments:

1- Please give full name of AIPP3 in either abstract or introduction.

2- Lines 557-559: please clarify statement. "...hypothesis that the BAH-PHD-CPL2 complex represses transcription through the inhibition of Ser5P-dependent transcription initiation but NOT (?) a histone ubiquitination-INDEPENDENT (?) mechanism".

3- The discussion section could be improved. It is clear that CPL2 is a plant specific protein and thus this process may be unique to plants. However, are there analogies in other systems? For example, previous findings linking histone readers to CTD phosphorylation? In addition, some comments about the general involvement of BPC in plant development could be included.

Point-to-point response to reviewers

Reviewer #1 (Remarks to the Author):

In this study, Zhang et al identify the BAH domain of AIPP3 and the PHD domain of AIPP2/PAIPP2 as the specific reader module of H3K27me3 and unmodified H3K4 in Arabidopsis, respectively; AIPP3 and AIPP2/PAIPP2 further associate with a plant-specific Pol2 carboxyl terminal domain (CTD) phosphatase CPL2 for targeted gene repression by altering Pol2 phosphorylation state. The authors verify that AIPP3 associates with AIPP2 and CPL2 to form a ternary complex (BAH-PHD-CPL2) in Arabidopsis, and genes in this complex is shown to act in the same pathway in flowering time control. They verify AIPP3-BAH domain as a H3K27me3 reader module by ITC and determine the crystal structure of BAH in complex with an H3K27me3 peptide. Likewise, PAIPP2-PHD is confirmed as a reader of unmodified H3K4; key residues involved in interaction are identified by structure modelling. They demonstrate that BAH-PHD-CPL2 represses the expression of H3K27me3-enriched genes, and that H3K27me3 binding is indispensable to transcription repression and flowering time control by structure-based mutagenesis in Arabidopsis. NET-seq and flavopiridol assay are performed to provide evidence that transcription of targeted gene transcription is repressed by BAH-PHD-CPL2. These data support a model that BAH-PHD-CPL2 bridges the cross-talk between H3K27me3 and Pol2 phosphorylation, which plays an important role in flowering time control in Arabidopsis.

This is a well-presented manuscript displaying a new mechanism of H3K27me3 mediated transcription regulation, which will be interesting to a broad audience. I recommend the acceptance of the manuscript for publication. Here are some modest questions and suggestions:

Question 1: The authors model the AIPP2 and PAIPP2 PHD fingers based on the structure of ATXR5 PHD finger to analyze the interaction between unmodified H3K4

and AIPP2/PAIPP2 PHD fingers, and identify several conserved residues which may involve in the recognition. I wonder if the author can provide further evidence to support their modelling result, such as ITC using the mutants.

Response:

Thanks very much for noticing us this important validation experiment. The AIPP2 protein is easy to precipitate and is not suitable for ITC. We only focus on the ITC with PAIPP2. PAIPP2 binds to the unmodified H3K4 peptide mainly through the main chain hydrogen bonds, while the side chain interaction is the H3R2 recognition by PAIPP2 Asp306. We found that D306K mutation almost totally aborted the peptide binding as shown by our ITC experiment. The new ITC data have been added into main figure 3 as a new panel Fig. 3I.

Question 2: Line 413-416: I would suggest the author providing a ChIP-qPCR result of AIPP3, W170A, Y149A/W170/Y172 in *aipp3-1* mutant strain to strengthen their conclusion that H3K27me3 by AIPP3 is indispensable for flowering time control and transcription repression.

Response:

Thanks for your precious suggestion. As suggested, ChIP-qPCR assays were performed in AIPP3 and W170A transgenic plants to examine AIPP3 binding on selected target genes. As shown in the following figure, compared to significant enrichment of AIPP3 in wild-type *AIPP3* transgene, enrichment of AIPP3 in W170A

transgene was abolished at the selected target genes *AGO5*, *SUC5* and *FT*, indicating that AIPP3 binding on selected target genes was disrupted by the W170A mutation. Considering W170 is required for H3K27me3 binding (Fig. 3h), this result strengthens our conclusion that H3K27me3 binding activity of AIPP3 is indispensable for flowering time control and transcriptional repression. Interestingly, we found that W170A and wild-type AIPP3 proteins have comparable accumulation in transgenic plants, whereas accumulation of transgenic W149A/W17A/Y172A protein is much lower than that of W170A and wild-type AIPP3, implying that W149A/W17A/Y172A protein is not stable in transgene. We have added the western blotting and ChIP-qPCR results to the new main figure as Fig. 6c and Fig. 6e.

Fig. 6c

Fig. 6e

In Line 413, I guess “Y149A” is a typo and should be “W170A”?

Response:

Thanks very much for pointing out our mistake. Yes, “Y149A” in line 413 is a typo of “W170A”. Now it has been corrected.

Question 3: Line 448: I wonder what is the level of Ser2P Pol2 in *aipp3-1*, *cpl2-2* and *aipp2-1/paipp2-1* mutants. It would be more informative to compare the fluctuation of Ser2P and Ser5P in the mutants since CPL2 was reported as a Pol CTD Ser5 phosphatase.

Response:

Thanks very much for your precious suggestion. As suggested, we performed ChIP-qPCR analysis of total (unphosphorylated) Pol II and Ser2P-Pol II in Col-0 and *bpc* mutants. As shown in the following figure, similar to Ser5P-Pol II, the occupancy of total and Ser2P-Pol II also displayed higher levels at selected target genes *AGO5*, *SUC5* and *FT* in the *bpc* mutants in comparison with Col-0. In contrast, the occupancies of all types of Pol II at *AGO5* and *SUC5* downstream non-target genes were not significantly changed, demonstrating that *bpc* mutations led to reactivation of Pol II initiation specifically at BPC complex target genes and the initiated Pol II successfully switched to an elongating state in the *bpc* mutants. The new ChIP-seq data has been added to supplementary data as new Supplementary Figure 12.

Other comments:

1. Line 327: “155 genes were commonly up-regulated by the BAH-PHD-CPL2 complex”. I guess it should be “in the bah-phd-cpl2 mutant”?

Response:

Thanks very much for pointing out the mistake. Now “by the BAH-PHD-CPL2 complex” has been changed to “in the bah-phd-cpl2 mutants”.

2. Fig5a left panel: There is a shift of genome site label “e, a, b, f”.

Response:

Thanks very much for pointing out this mistake. Now the labeled have been adjusted.

3. Line 384 and Fig 5e: "... AIPP3-enriched loci, significantly overlap ...” p-value should be provided as the authors mention “significantly overlap”.

Response:

Thanks for your precious suggestion. We performed Fisher’s Exact Test for the overlap analysis. The results show that the p-values of all mutual overlaps are less than $2.2204e-16$, indicating that all the overlaps are significant. Now the p-values have been added to Fig. 5f.

Reviewer #2 (Remarks to the Author):

This is an interesting study that progresses our understanding of mechanisms of how chromatin modifications influence transcription. The authors characterize an H3K27me3 reader, first using a peptide pull-down assay and then ITC. They continue by defining the BAH structure by co-crystallizing the BAH domain with H3K27me3. They corroborate and validate the histone recognition with a mutation analysis to show how the binding is affected after mutation of the aromatic cavity that provides an optimal accommodation of K27me3.

They also analyse the PHD fingers PAIPP2 and AIPP2-PHD. Alignments and structural predictions for the PHDs indicate a possible interaction with unmodified H3K4 and this was confirmed for PAIPP2-PHD by ITC.

Specific comments:

Question 1: Lines 218-220: the ITC graph they refer to in Fig. 3 c shows K_d values that match the sequential increase, however the curves in this graph do not correspond to the K_d s – is there a mixup in the colouring of dark blue and light blue in the legend?

Response:

Thanks for pointing out this mislabeling. We have relabeled the curves.

Question 2: lines 388-393 and lines 555-559: in the results section the authors make a connection of their complex to PRC1, but they only show that many AIPP3-enriched loci overlap with EMF1-bound loci. Since both associate with H3K27me3 this could just be a correlation. If EMF1 is found in the IP-MS dataset for AIPP3, this would make the claims more convincing.

Response:

Thanks for your question. We agree that the overlap of EMF1-bound loci and AIPP3-enriched loci may be due to their association with H3K27me3. As suggested, we searched the co-purified protein with BPC complex but no EMF1 was found (new Supplementary Table 1). One possible explanation for this result is that AIPP3 and EMF1 as H3K27me3 reader proteins may be able to recognize same loci but function independently in mediating transcriptional repression, although a possibility of indirect association between AIPP3 and EMF1 cannot be excluded. Now we have rephrased this section as follows:

“While, no EMF1 peptides were found in the BPC complex co-purified proteins (Supplementary Table 1). One possible explanation is that both EMF1 and AIPP3 associate with H3K27me3 mark but function independently in mediating downstream transcriptional repression, although the possibility of indirect association between these two reader proteins cannot be excluded.”

The double negative “not a histone ubiquitination-independent mechanism” in line 559 needs rephrasing.

Response:

Thanks for pointing out this mistake. Now this section has been rephrased.

Showing global levels of H2Aub is not enough to support a broad statement on dependency of H2Aub (eg. they might only see differences when they look at their specific target genes), so this section of the text needs to be carefully edited.

Response:

Thanks for your precious question. We agree with your comment that global levels of H2Aub is not enough to support a broad statement on dependency of H2Aub. We rephrased this section as follows:

“We found that the H2Aub1 levels were reduced in the *lhp1-3* mutant but were not affected in the *aipp3-1/cpl2-2*, *aipp3-1/aipp2-1/paipp2-1* and *aipp2-1/paipp2-1/cpl2-2* mutants (Supplementary Fig. 10a). This observation is consistent with our hypothesis that the BPC complex represses transcription mainly through the inhibition of Ser5P-dependent transcription initiation. While, we cannot rule out the possibility that the BPC complex has direct/indirect interaction with H2Aub1 at specific target genes.”

Question 3: lines 409-421: in connection to H3K27me3 binding of AIPP3, they introduce mutated AIPP3 transgenes (W170A, Y149A/W170A, Y172A) into the *aipp3-1* mutant and say that the mutated transgenes do not complement the flowering time phenotype and the repressive state of the selected target genes. However, this would also be true for transgenes that do not produce a stable protein independent of any binding change. A western showing that protein levels of AIPP3 are restored in mutated AIPP3 transgenic lines is essential and ideally also a ChIP experiment showing that chromatin binding of mutated AIPP3 is lost at selected target genes as a more direct evidence for in vivo H3K27me3 binding.

Response:

Thanks for your precious question. As we mentioned in the response to reviewer 1, we examined protein accumulation levels in different transgenes by Western blotting. The result indicated that AIPP3 and W170A proteins have comparable accumulation levels in transgenes, whereas Y149A/W170A/Y172A protein is not stably expressed in transgene. Therefore, we only used AIPP3 and W170A transgenes in the other experiments. The Western blotting result has been added to new Fig. 6 as a panel Fig. 6c.

As suggested, we also performed ChIP-qPCR assay in Col-0, *AIPP3* and *W170A*

transgenic plants. The result indicated that W170A mutation makes AIPP3 lose its chromatin binding at selected target genes. This result strengthens our conclusion that H3K27me3 is indispensable for AIPP3-mediated transcriptional repression and flowering time control. Now we have added this result to new Fig. 6 as a panel Fig. 6e. This section has been rephrased in the revision.

Fig. 6c

Fig. 6e

Question 4: Lines 452-466, on coupling with Pol II release: to support the coupling hypothesis they rely on FLA treatments based on the previous report that CPL2 dephosphorylates CTD Ser5. However, this experiment cannot be used to directly link BAH-PHD-CLP2 complex activity to phosphorylation of Pol II, because of the likelihood of general, indirect effects on transcription at a global level.

Response:

Thanks for your precious comment. We agree with you that FLA treatment experiment is not a direct evidence to link BPC complex to phosphorylation of Pol II. To strengthen our hypothesis, we compared the chromatin binding of CPL2 in the presence or absence of AIPP3 and PHD proteins. To this end, *CPL2-GFP* transgene was crossed into *aipp3-1* and *aipp2-1/paipp2-1* mutants and CPL2 ChIP-qPCR assay was performed in *CPL2-GFP/Col-0*, *CPL2-GFP/aipp3-1* and *CPL2-GFP/aipp2-1/paipp2-1* plants. The result indicated that CPL2 has significant

binding at selected target genes in wild-type background, whereas this binding was completely abolished in the *aipp3-1* and *aipp2-1/paipp2-1* mutants. This result provides a link between chromatin marks and CPL2-mediated dephosphorylation of Pol II, in which the chromatin localization of CPL2 at BPC complex target genes largely depends on the recognition of H3K27me3/H3K4me0 marks by BAH-PHD proteins. Now this result has been added to new Fig. 7d. This section has been rephrased in the manuscript revision.

Minor issues:

-the introduction in particular needs editing for English

Response:

Thanks for your suggestion. Now we have carefully revised the manuscript and improved the language by a native English editor in the revision.

-there is no reference to datasets in the text - full datasets for IP-MS (Fig. 1), RNAseq (Fig. 4) and ChIP-seq of AIPP3 (Fig. 5) should be provided.

Response:

Thanks very much for your reminder. Now the IP-MS result has been added as Supplementary Table 1. mRNA-seq and AIPP3 ChIP-seq datasets have been deposited in the GEO with the accession codes GSE147981 and GSE157196, respectively.

- line 161-163, not all phenotypic developmental defects described for *aipp3*, *cpl2* and *aipp2/paipp2* mutants are visible from the images shown (Fig. 2a)

Response:

Thanks very much for your suggestion. Now more phenotypic developmental defects have been added as new Fig. 2a.

-line 247, the reference to Supplementary Fig. 3b should be Suppl. Fig. 7

Response:

Thanks very much for pointing out it. We are sorry for this mistake. Now “Supplementary Fig. 3b” has been changed to “Supplementary Fig. 7”

-For Fig. 4b, does the calculated FPKM percentage only refer to up-DEGs in the mutants? This is not clear from the figure legend.

Response:

Thanks for your question. Yes, the calculated FPKM percentage only refer to up-DEGs in the mutants. Now we have rephrased the figure legend.

- In the supplementary Fig.8a AtPHD2-PHD should be changed to AtPAIPP2-PHD.

Response:

Thanks very much for pointing out this mistake. Now “AtPHD2-PHD” has been changed to “AtPAIPP2-PHD”.

Reviewer #3 (Remarks to the Author):

In this manuscript, Zhang and colleagues report the functional characterisation of the Arabidopsis BAH-PHD-CPL2 complex (BPC complex), defining its role in transcriptional gene silencing. BPC is composed by the PHD-domain proteins AIPP2/PAIPP2, the BAH-domain protein AIPP3 and the Pol CTD Ser5 phosphate CPL2. Each complex component has a differential activity: 1) recognition of the PRC2 repressive histone modification H3K37me3 by BAH domains, 2) recognition of H3K4me3 by PHD domains, 3) Pol II CTD dephosphorylation by CPL2. By conducting different protein-interaction analyses (including mass-spectrometry, Y2H, luciferase split assay, gel filtration followed by immunoblot analysis), the authors

very robustly showed the interaction of AIPP2 and PAIPP2 with AIPP3 and CPL2 proteins. Based on the data, this work proposes the existence of two types of BPC complexes, AIPP2/AIPP3/CPL2 and PAIPP2/AIPP3/CPL2, with AIPP2 and PAIPP2 having redundant functions. The authors confirmed the capacity of AIPP3-BAH domain to recognize H3K27me3 using different approaches, including structural studies. Additionally, this work demonstrates the capacity of AIPP2- and PAIPP2-PHD domains to bind to unmethylated H3K4.

BPC complexes seem to play a very important role in Arabidopsis development. The *aipp3* and *cpl2* single mutants, as well as the *aipp2/paipp2* double mutants, are early flowering both in long and short days. The early flowering phenotype correlates with increased levels of FT mRNA in *aipp3*, *cpl2* and *aipp2/paipp2*, and was rescued by the *ft-10* mutation. In addition, combinations of triple mutant analyses showed similar phenotypes as those of the single mutants, demonstrating that the BPC complex components belong to the same genetic pathway.

RNA-seq and ChIP-qPCR assays revealed a connection of BPC with gene repression, as well as a link to PRC2-mediated silencing. Furthermore, the authors demonstrated that AIPP3 target genes are enriched with the H3K27me3 repressive mark, whereas H3K4me3 levels are reduced at those genes. Interestingly, the absence of BPC leads to accumulation of the phosphorylated version of RNA Pol II that is involved in productive transcription elongation (Ser5P Pol II) over two selected BPC target genes. Therefore, this work documents that the combined activities of the three BPC complex components result in coupling of H3K27me3 recognition with direct regulatory modifications of RNA Pol II, eventually leading to transcriptional repression. This manuscript proposing a direct connection of H3K27me3 to inhibiting RNA Pol II elongation represents an interesting advance to the PRC2-mediated gene silencing mechanism in plants. I have some comments that should be addressed to improve this manuscript.

Major points:

Question 1: Data presented in this work suggest that *FT* is an important target of BPC complex, explaining the early flowering phenotype of the *aipp3*, *cpl2* and *aipp2/paipp2* mutants. Is *FT* a direct target of BPC? Including *FT* locus in the ChIP-seq analysis shown in Figure 5A would help to clarify this point. Similarly, it would be good to include *FT* locus in Figure 6B-C.

Response:

Thanks very much for this good question. To answer this question, we first examined the ChIP-seq result of AIPP3. Unfortunately, we did not observe obvious enrichment of AIPP3 at *FT* locus. To further clarify this point, we performed independent ChIP-qPCR of AIPP3, AIPP2 and PAIPP2 with three biological repeats. As shown in Fig. 5a, the results indicated that both AIPP3, AIPP2 and PAIPP2 have significant binding at *FT* locus. Moreover, we also compared AIPP3's binding at *FT* in wild-type and W170A transgenic plants by performing ChIP-qPCR assay. The results indicated that the W170A mutation of AIPP3 abolished its binding at *FT* locus (new Fig. 6e), suggesting the H3K27me3 reader activity is indispensable for AIPP3's binding at this locus. In addition, as shown in Fig. 7d, chromatin binding analysis via ChIP-qPCR indicated that CPL2 has significant binding at *FT* locus and this binding was disrupted in *aipp3* and *aipp2/paipp2* mutants, suggesting that BAH-PHD module-mediated recognition of H3K27me3/H3K4me0 is required for CPL2's occupancy at *FT*. Based on these evidence, we think *FT* should be a direct target of BPC complex. We speculate that the low enrichment of AIPP3 in ChIP-seq data may be due to many factors. One possible factor is the weak transcriptional activity of *FT* in Col-0 background which can be verified by the low Pol II occupancy (Fig. 7a). The other possible factor is the weak binding of AIPP3 at *FT* which may lead to a failure in the detection of ChIP-seq signal. Consistent with this notion, AIPP3 occupancy at *FT* locus is obviously lower than *AGO5* and *SUC5* (Fig. 5a). Of course, we cannot rule out the possibility that BPC complex also has an indirect regulation on *FT* expression. We have added the data to new Figures and rephrased the corresponding parts in results and discussion.

Fig. 5a. ChIP-qPCR of AIPP3, AIPP2 and PAIPP2 at *FT* locus

Fig. 6e. W170A mutation of AIPP3 abolishes its binding at *FT* locus

Fig. 7d. CPL2 binds to the chromatin of *FT*

Question 2: As the authors claim that BPC is driving transcriptional repression, it would be more suitable to measure the levels of nascent (unspliced) RNAs in Figure 6C.

Response:

Thanks very much for this precious suggestion. As suggested by the reviewer, we performed Nuclear Run-On assays to measure the levels of nascent RNAs in different mutants with or without FLA treatments. As shown in new Fig. 7c, the nascent RNA levels of selected target genes, including *FT*, but not that of the downstream non-target genes, were dramatically increased in the *bpc* mutants compared to Col-0 in mock condition, strongly supporting our conclusion that BPC implements transcriptional repression. In contrast, after FLA treatment, the nascent RNA levels

were reduced, and no significant changes were observed between different genotypes. Now this section has been rephrased in the revision.

Question 3: To fully support the idea of Ser5 CTD dephosphorylation by the CPL2 module of BPC, I would suggest that the authors test enrichment of total Pol II (unphosphorylated) and Ser2 by ChIP-qPCR, comparing Col0 vs *bpc* mutants. In addition, in the analysis of Figure 6B, including primers over the downstream genes (AT2G27900 and AT1G71900) would help to define whether the differences shown are significant.

Response:

Thanks very much for your question. This is a good suggestion. As we mentioned in the response to reviewer 1, we performed ChIP-qPCR analysis of total (unphosphorylated) Pol II and Ser2P-Pol II in Col-0 and *bpc* mutants. As shown in the following figure, similar to Ser5P-Pol II, the occupancy of total and Ser2P-Pol II also displayed higher levels at selected target genes *AGO5*, *SUC5* and *FT* in the *bpc* mutants in comparison with Col-0. In contrast, the occupancies of all types of Pol II at *AGO5* and *SUC5* downstream non-target genes were not significantly changed, demonstrating that *bpc* mutations led to reactivation of Pol II initiation specifically at BPC complex target genes and the initiated Pol II successfully switched to an elongating state in the *bpc* mutants. The new ChIP-seq data has been added to supplementary data as new Supplementary Figure 12.

Question 4: I think that the general message and model proposed (Figure 6D) in this work could be further improved. Currently, a general repression mechanism is proposed, linking PRC2 repressive activity to preventing RNA Pol II productive elongation by the action of CPL2. In my view, the function of BPC is preventing Pol II elongation in loci that were already silenced by PRC2 (were already covered by H3K27me3). In other words, a surveillance system to prevent gene reactivation. This could be supported by the fact that no changes in H3K27me3 were observed in the selected genes in the different BPC mutants analysed (Figure 4G), and yet mRNA levels increase (Figure 4F). In this regard, testing H3K27me3 enrichment over more

BPC targets in the different mutants would strengthen this work. Also, and similarly to that stated above, it would be good to provide data for nascent transcripts.

Response:

Thanks very much for the precious suggestion. As suggested, we selected more BPC target genes to compare the difference of H3K27me3 deposition between WT and *bpc* mutants, and measure nascent RNA levels via Nuclear Run-On method. As shown in new Supplemental Fig. 10b-d, three target genes showing AIPP3 binding, *AT2G43570*, *AT3G59480* and *AT3G53650*, were selected. The Nuclear Run-On results showed the nascent RNA levels of all these three genes were greatly increased in *bpc* mutants compared to Col-0, supporting our conclusion that BPC complex confers transcriptional repression at these genes. Interestingly, regarding H3K27me3 deposition, ChIP-qPCR results indicated that H3K27me3 levels were not obviously changed in *bpc* mutants at two target genes, *AT2G43570* and *AT3G59480*, consistent with our observation at *AGO5*, *SUC5* and *FT* (Fig. 4g). While, H3K27me3 deposition showed significant reduction at *AT3G53650*, implying that BPC dysfunction has substantial impacts on H3K27me3 deposition at this gene. Combined with these data, we speculate that, as the reviewer pointed out, for most target genes, the BPC complex may serve as a surveillance system to prevent reactivation of H3K27me3-marked genes which are already silenced by PRC2, but in some specific genes, BPC is required for H3K27me3 through unknown mechanism. Now we have rephrased this section. Based on these evidence, we also revised our model as a new Fig. 8.

Supplementary Fig. 10 The effects of BPC dysfunction on the deposition of different histone marks.

Fig. 8 A working model of the BPC complex-mediated transcription repression

Minor comments:

1- Please give full name of AIPP3 in either abstract or introduction.

Response:

Thanks very much for pointing out our negligence. We are sorry for that. Now the full name of AIPP3 have been added in introduction.

2- Lines 557-559: please clarify statement. "...hypothesis that the BAH-PHD-CPL2 complex represses transcription through the inhibition of Ser5P-dependent transcription initiation but NOT (?) a histone ubiquitination-INDEPENDENT (?) mechanism".

Response:

Thanks for pointing out our negligence. Now this section has been rephrased.

3- The discussion section could be improved. It is clear that CPL2 is a plant specific protein and thus this process may be unique to plants. However, are there analogies in other systems? For example, previous findings linking histone readers to CTD phosphorylation? In addition, some comments about the general involvement of BPC in plant development could be included.

Response:

Thanks very much for this good suggestion. Pol II CTD phosphatases are conserved in eukaryotes. Although it is a well-known knowledge that CTD phosphatase can affect transcription through modulation of Pol II CTD phosphorylation, no evidence shows that histone modifiers manipulate CTD phosphatases, including CPL2 analogues in other systems, to affect transcription. Regarding histone readers, several studies have reported that histone PTM modifiers can affect transcription via modulating Pol II CTD phosphorylation state (Kooistra and Helin, 2012; Srivastava and Ahn, 2015). JMJD3, a H3K27me3 demethylase in human, has been shown to directly interact with CTD-Ser2P to affect gene expression (Estarras et al., 2013). Knocking down JMJD3 or JHDM1D, a H3K27me1/2 demethylase, reduces the enrichment of Pol II CTD-Ser2P at specific genes in human promyelocytic leukemia

cells (Chen et al., 2012). In addition to histone methylation, other histone PTMs including histone ubiquitylation and phosphorylation, are also associated with Pol II CTD phosphorylation-dependent transcriptional elongation (Srivastava and Ahn, 2015). For example, knocking down histone ubiquitylation modifiers has been shown to affect CTD-Ser2 phosphorylation (Srivastava and Ahn, 2015). The phosphorylation of histone H3 on S10 and S28 has been reported to be associated with phosphorylated Pol II during transcriptional activation in humans and *Drosophila* (Ivaldi et al., 2007; Rossetto et al., 2012). These studies support a notion that modulation of Pol II CTD phosphorylation represents an important regulatory mechanism adopted by chromatin regulators to regulate gene expression. Now we have added these discussions in the manuscript revision.

As suggested, we also added some comments on the general involvement of BPC in plant development.

Chen, S., Ma, J., Wu, F., Xiong, L.J., Ma, H., Xu, W., Lv, R., Li, X., Villen, J., Gygi, S.P., *et al.* (2012). The histone H3 Lys 27 demethylase JMJD3 regulates gene expression by impacting transcriptional elongation. *Genes & development* 26, 1364-1375.

Estaras, C., Fueyo, R., Akizu, N., Beltran, S., and Martinez-Balbas, M.A. (2013). RNA polymerase II progression through H3K27me3-enriched gene bodies requires JMJD3 histone demethylase. *Molecular biology of the cell* 24, 351-360.

Ivaldi, M.S., Karam, C.S., and Corces, V.G. (2007). Phosphorylation of histone H3 at Ser10 facilitates RNA polymerase II release from promoter-proximal pausing in *Drosophila*. *Genes & development* 21, 2818-2831.

Kooistra, S.M., and Helin, K. (2012). Molecular mechanisms and potential functions of histone demethylases. *Nature reviews Molecular cell biology* 13, 297-311.

Rossetto, D., Avvakumov, N., and Cote, J. (2012). Histone phosphorylation: a chromatin modification involved in diverse nuclear events. *Epigenetics* 7, 1098-1108.

Srivastava, R., and Ahn, S.H. (2015). Modifications of RNA polymerase II CTD: Connections to the histone code and cellular function. *Biotechnology advances* 33, 856-872.

REVIEWERS' COMMENTS

Reviewer #1 (Remarks to the Author):

My concerns have been well addressed with the new ITC and ChIP-qPCR data. Overall this is a conceptually important study that revealed a new molecular mechanism on H3K27me3-mediated gene silencing. I strongly support its publication.

Fig. 8, "Pi" (inorganic phosphate) should be kept in one line.

Reviewer #2 (Remarks to the Author):

The authors have done a good job at addressing the reviewers comments. The structural analysis of the AIPP3-BAH domain and PAIPP2-PHD finger is an important contribution to the field. The functional analysis of the role of the interacting CPL2 is less clear-cut due to non-linearity and feedback mechanisms complicating interpretation. However, this work will be of considerable widespread interest.

The manuscript still needs considerable editing - for example, there is a stray 'and' in the abstract.

Reviewer #3 (Remarks to the Author):

This manuscript have significantly improved. The authors have answered all my questions and revised the manuscript accordingly. I have no further comments.

Point-by-point response

Reviewer #1 (Remarks to the Author):

My concerns have been well addressed with the new ITC and ChIP-qPCR data. Overall this is a conceptually important study that revealed a new molecular mechanism on H3K27me3-mediated gene silencing. I strongly support its publication.

Fig. 8, "Pi" (inorganic phosphate) should be kept in one line.

Response: We thank the reviewer's professional comment. Now the error has been corrected.

Reviewer #2 (Remarks to the Author):

The authors have done a good job at addressing the reviewers comments. The structural analysis of the AIPP3-BAH domain and PAIPP2-PHD finger is an important contribution to the field. The functional analysis of the role of the interacting CPL2 is less clear-cut due to non-linearity and feedback mechanisms complicating interpretation. However, this work will be of considerable widespread interest.

The manuscript still needs considerable editing - for example, there is a stray 'and' in the abstract.

Response: We thank the reviewer's precious suggestion and helpful criticism. Now we have carefully revised the manuscript.

Reviewer #3 (Remarks to the Author):

This manuscript have significantly improved. The authors have answered all my questions and revised the manuscript accordingly. I have no further comments.

Response: We thank the reviewer's comment.